# Reinforcement biases subsequent perceptual decisions when confidence is low, a widespread behavioral phenomenon

Armin Lak[1,2]\*, Emily Hueske[3,4,5], Junya Hirokawa[6,7], Paul Masset[3,6,8], Torben Ott[6,9], Anne E Urai[6,10], Tobias H Donner[10], Matteo Carandini[2], Susumu Tonegawa[4,11], Naoshige Uchida[3], Adam Kepecs[6,9]\*

[1]Department of Physiology, Anatomy and Genetics, University of Oxford, Oxford, United Kingdom; [2]UCL Institute of Ophthalmology, University College London, London, United Kingdom; [3]Department of Molecular and Cellular Biology and Center for Brain Science, Harvard University, Cambridge, United States; [4]RIKEN-MIT Laboratory at the Picower Institute for Learning and Memory at Department of Biology and Department of Brain and Cognitive Science, Massachusetts Institute of Technology, Cambridge, United States; [5]McGovern Institute for Brain Research at Department of Brain and Cognitive Sciences, Massachusetts Institute of Technology, Cambridge, United States; [6]Cold Spring Harbor Laboratory, Cold Spring Harbor, United States; [7]Graduate School of Brain Science, Doshisha University, Kyotanabe, Kyoto, Japan; [8]Watson School of Biological Sciences, Cold Spring Harbor, United States; [9]Departments of Neuroscience and Psychiatry, Washington University School of Medicine, St. Louis, United States; [10]Department of Neurophysiology, University Medical Center, Hamburg-Eppendorf, Hamburg, Germany; [11]Howard Hughes Medical Institute at Massachusetts Institute of Technology, Cambridge, United States

**\*For correspondence:**
armin.lak@dpag.ox.ac.uk (AL);
akepecs@wustl.edu (AK)

**Abstract** Learning from successes and failures often improves the quality of subsequent decisions. Past outcomes, however, should not influence purely perceptual decisions after task acquisition is complete since these are designed so that only sensory evidence determines the correct choice. Yet, numerous studies report that outcomes can bias perceptual decisions, causing spurious changes in choice behavior without improving accuracy. Here we show that the effects of reward on perceptual decisions are principled: past rewards bias future choices specifically when previous choice was difficult and hence decision confidence was low. We identified this phenomenon in six datasets from four laboratories, across mice, rats, and humans, and sensory modalities from olfaction and audition to vision. We show that this choice-updating strategy can be explained by reinforcement learning models incorporating statistical decision confidence into their teaching signals. Thus, reinforcement learning mechanisms are continually engaged to produce systematic adjustments of choices even in well-learned perceptual decisions in order to optimize behavior in an uncertain world.

## Introduction

Learning from the outcomes of decisions can improve subsequent decisions and yield greater success. For instance, to find the best meal on a busy street where restaurants often change menus,

one needs to frequently sample food and learn. Humans and other animals efficiently learn from past rewards and choose actions that have recently lead to the best rewards (*Daw et al., 2006*; *Lee et al., 2012*; *Samejima et al., 2005*; *Tai et al., 2012*). In addition to evaluating past rewards, decision making often require consideration of present perceptual signals; the restaurants' signs along the busy street might be too far and faded to be trusted. Therefore, good decisions ought to take into account both current sensory evidence as well as the prior history of successes and failures.

Decisions guided by the history of rewards can be studied in a reinforcement learning framework (*Sutton and Barto, 1998*). Perceptual decisions, on the other hand, have been classically conceptualized within a statistical, psychometric framework (*Green and Swets, 1966*). Although statistical decision theory and reinforcement learning provide two largely distinct frameworks for studying decisions, we are often challenged by both limits in our perception as well as limits in learning from past rewards. For sensory decisions, classical psychometric analysis estimates three fundamental variables that determine the quality of choices: the bias, lapse rate and sensitivity (*Green and Swets, 1966*; *Wichmann and Hill, 2001*). When bias and lapse rates are negligible and sensitivity has reached its maximum over time, then fluctuations in decisions are solely attributed to the noise in the perceptual processing. Under these assumptions, incorrect decisions are caused by perceptual noise creating imperfect percepts. Here, we show a systematic deviation from the assumption of no learning during well-trained perceptual decisions: past rewards bias perceptual choices specifically when the previous stimulus was difficult to judge, and the confidence in obtaining the reward was low.

In laboratory perceptual decision-making paradigms, there is typically no overt learning after the task acquisition is complete. Nevertheless, several studies have shown that past rewards, actions, and stimuli can appreciably influence subsequent perceptual choices (*Abrahamyan et al., 2016*; *Akaishi et al., 2014*; *Akrami et al., 2018*; *Braun et al., 2018*; *Busse et al., 2011*; *Cho et al., 2002*; *Fan et al., 2018*; *Fischer and Whitney, 2014*; *Fritsche et al., 2017*; *Fründ et al., 2014*; *Gold et al., 2008*; *Hwang et al., 2017*; *Lueckmann et al., 2018*; *Luu and Stocker, 2018*; *Marcos et al., 2013*; *Tsunada et al., 2019*; *Urai et al., 2017*). Some of these observations support the view that simple forms of reward-based learning are at work during asymptotic perceptual performance. For instance, subjects might repeat the previously rewarded choice or avoid it after an unsuccessful trial (*Abrahamyan et al., 2016*; *Busse et al., 2011*; *Tsunada et al., 2019*; *Urai et al., 2017*). However, these types of choices biases seem to be suboptimal and might reflect simple heuristics. Thus, the extent to which choice biases in perceptual decisions can be expected from normative considerations in reinforcement learning has been unclear. Perhaps, the most prominent prediction of reinforcement learning under perceptual uncertainty is that the strength of sensory evidence (i.e. confidence in the accuracy of a decision) should modulate how much to learn from the outcome of a decision (*Lak et al., 2017*; *Lak et al., 2019*). Outcomes of easy decisions are highly predictable, and thus there is little to be learned from such decisions. In contrast, outcomes of difficult, low confidence decisions, provide the most prominent opportunity to learn and adjust subsequent decisions (*Lak et al., 2017*; *Lak et al., 2019*). These considerations lead to the hypothesis that decision confidence regulates trial-by-trial biases in perceptual choices.

Here, we demonstrate that well-trained perceptual decisions can be systematically biased based on previous outcomes in addition to current sensory evidence. We show that these outcome-dependent biases depend on the strength of past sensory evidence, suggesting that they are consequences of confidence-guided updating of choice strategy. We demonstrate that this form of choice updating is a widespread behavioral phenomenon that can be observed across various perceptual decision-making paradigms in mice, rats and humans. This trial-to-trial choice bias was also present in different sensory modalities and transferred across modalities in an interleaved auditory/olfactory choice task. To explain these observations, we present a class of reinforcement learning models and Bayesian classifiers that adjust learning based on the statistical confidence in the accuracy of previous decisions.

# Results

## Perceptual decisions are systematically updated by past rewards and past sensory stimuli

To investigate how the history of rewards and stimuli influences subsequent perceptual decisions, we began with an olfactory decision task (*Figure 1a*). Rats were trained on a two-alternative choice olfactory decision task (*Uchida and Mainen, 2003*). Two primary odors were associated with rewards at left and right choice ports and mixtures (morphs) of these odors were rewarded according to a categorical boundary (50:50 mixture; *Figure 1a*). To manipulate perceptual uncertainty, we varied the odor mixtures, that is the ratio of odor A and B in a trial-by-trial manner, testing mixtures 100:0, 80:20, 65:35, 55:45, 45:55, 35:65, 20:80 and 0:100. Rats showed near-perfect performance for easy mixtures and made errors more frequently for difficult mixtures (*Figure 1a*, bottom panel).

After task learning, rats showed stable behavior across testing sessions (*Figure 1b*). To quantify behavioral performance and stability across sessions, we fitted choice behavior with psychometric

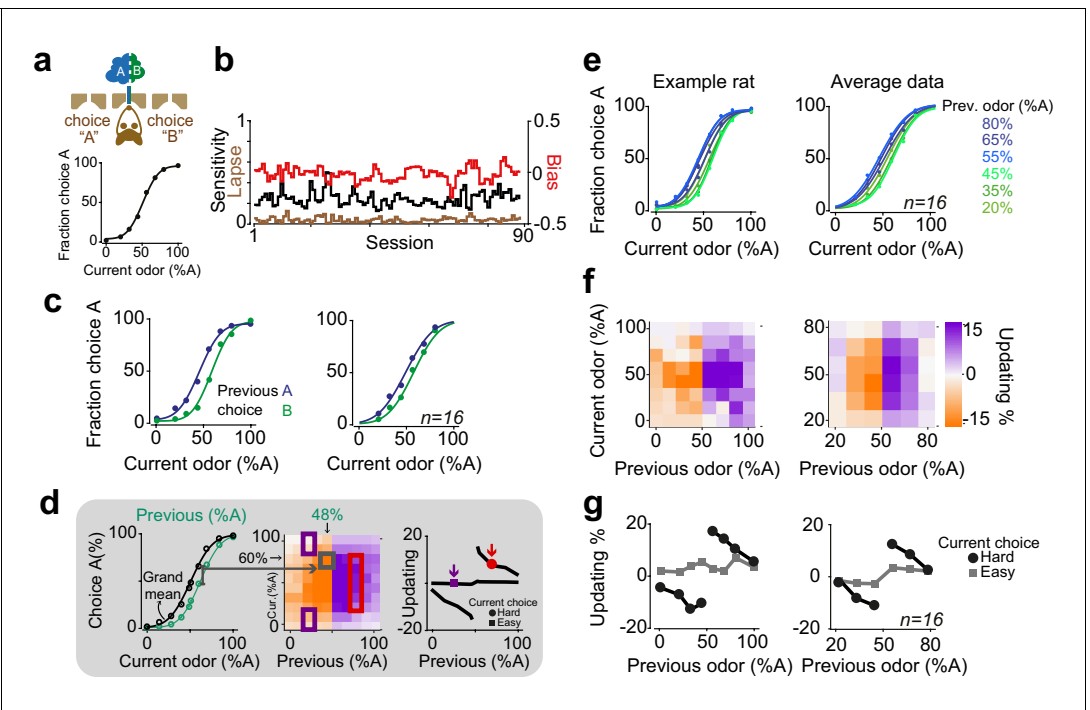

**Figure 1.** Rats update their trial-by-trial perceptual choice strategy in a stimulus-dependent manner. (**a**) Top: Schematic of a 2AFC olfactory decision-making task for rats. Bottom) Average performance of an example rat. (**b**) Following learning, the psychometric curves showed minimal fluctuations across test sessions. Bias, sensitivity and lapse were measured for each test session. (**c**) After successful completion of a trial, rats tended to shift their choice toward the previously rewarded side. Left and right panels illustrate example animal and population average. (**d**) Schematic of analysis procedure for computing conditional psychometric curves and updating plots. Left: Black curve shows the overall psychometric curve and the green curve shows the curve only after trials with 48% odor A (i.e. conditional on the stimulus (48% A) in the previous trial). Middle: Each point in the heatmap indicates the vertical difference between data points of the conditional psychometric curve and the overall psychometric curve. Red and purple boxes indicate data points which are averaged to compute data points shown in the rightmost plot. Right: Updating averaged across current easy trials (in this case the easiest two stimulus levels) and current difficult trials. (**e**) Performance of the example rat (left) and population (right) computed separately based on the quality of olfactory stimulus (shown as colors mixtures from blue to green) in the previously rewarded trial. After successful completion of a trial, rats tended to shift their choices towards the previously rewarded side but only when the previous trial was difficult. (**f**) Choice updating, that is the size of shift of psychometric curve relative to the average psychometric curve, as a function of sensory evidence in the previously rewarded trial, and current trial. Positive numbers refer to a bias towards choice A and negative numbers refer to a bias toward the alternative choice. The left and right plots refer to the example rat and population, respectively. (**g**) Choice updating as a function of previous stimulus separated for current easy (square) and difficult (circle) trials. These plots are representing averages across graphs presented in f.

The online version of this article includes the following figure supplement(s) for figure 1:

**Figure supplement 1.** Left: Performance of population of rats (n=16) computed from trials in which the previous stimulus was difficult (45% odor A, 55% odor B), separated based on whether the previous choice was rewarded (correct) or unrewarded (error).

functions that included parameters for sensitivity (reflecting perceptual noise), bias (the tendency to take one specific action) and lapse rate (stimulus-independent occasional errors possibly reflecting attentional or learning deficits) (*Figure 1a, b*). We found that in well-trained rats, the bias was near zero (2±4.6 %Odor A (mean± S.D), p = 0.08, signed rank test). Likewise, lapse rates were low (3±4%), indicating that for easy stimuli rats' performance was near perfect and not substantially degraded by attention or incomplete learning. Lapse rate, sensitivity and bias remained stable over sessions, indicating that rats reached asymptotic performance (*Figure 1b*, 14/16 rats, p > 0.1, linear regression).

Despite the stable, asymptotic performance for easy decisions, previous trials had substantial effects on subsequent choices (*Figure 1c*). In order to assess the effects of reinforcement on perceptual decisions, we calculated conditional psychometric functions. We first considered the effect of previous choice (*Figure 1c*). Psychometric functions were systematically biased by previously rewarded choices: after correct leftward choices rats tended to make a left choice. Conversely, following correct rightward choices, animals made rightward choices more often (F = 29.8, p=0.001, 2-way ANOVA).

The effects of the previous decision on subsequent choices also depended on the difficulty of the previous choices (*Figure 1d-g*). We computed psychometric functions after correct (and hence rewarded) trials separately for different stimuli of the previous trial (*Figure 1d*). The resulting psychometric functions were systematically biased towards the recently rewarded side for difficult decisions (*Figure 1e*). Rats tended to repeat their previous choices particularly when they succeeded to correctly categorize a challenging odor mixture and earn reward (*Figure 1e*). We quantified the magnitude of this choice bias for each pair of current and previous stimuli (*Figure 1d*, see Materials and methods). To do so, we subtracted the average psychometric curve (computed from all trials) from each psychometric curve computed conditional on the specific previous correct stimulus, and plotted the size and sign of this difference (positive: bias to choose A; negative: bias to choose B) (*Figure 1d,f*). To summarize choice biases, we then averaged these differences across trials in which the current choice was easy or difficult (*Figure 1d,g*). The magnitude of this choice bias was proportional to the difficulty of the previous decision, in addition to the difficulty of the current decision (*Figure 1f*). Updating was minimal when the current stimulus was easy, regardless of the difficulty of the previous decision (*Figure 1g*, squares). This is because the data points are overlapping when the current stimulus is easy, and hence the distance between them is close to zero (*Figure 1d, e*). When the current stimulus was difficult, updating was also minimal after correct easy choices, whereas it was strong following correct difficult choices (*Figure 1g*, circles, p=0.0002, rank sum test). Thus, when the current sensory evidence was strong, it determined the choice, without detectable effects of the previous trial (*Figure 1g*, squares). However, when the sensory evidence in the current trial was weak, the previous reward influenced choices only if the reward was earned in a difficult trial (*Figure 1g*, circles). The difficulty of previous decision did not influence the slope (sensitivity) nor the lapse of psychometric curves in the next trial (p>0.1 rank sum test). Additionally, plotting the psychometric curves conditional on the specific stimulus in the previous trial but separated according to the outcome of the previous trial (correct vs error) further illustrated that past outcome influence subsequent choice in particular when the previous choice was difficult (*Figure 1—figure supplement 1*). Together, these observations indicate that the effects of past rewards on perceptual choices depend on the difficulty of the previous perceptual judgments.

## Choice updating is not due to slow drifts in choice side bias

The results so far demonstrate that previous rewards influence subsequent perceptual decisions especially when the previous decision was difficult. One possibility is that these behavioral effects arise as a byproduct of slow fluctuations of side bias causing correlations across consecutive trials and hence systematic shifts in choices. This scenario can be illustrated within a signal detection theory (SDT) framework. In SDT, the perceived stimulus is compared to a decision boundary and produces a correct choice when the stimulus falls on the appropriate side of the boundary (*Figure 2a*). When the decision boundary is fixed, there is no apparent updating, as expected (*Figure 2—figure supplement 1a, b*). However, simulating a slowly drifting decision boundary reveals systematic effects of previous trials on next choices, because the drifting boundary induces correlations across trials, producing apparent choice updating based on the previous trials (*Figure 2b*; *Figure 2—figure supplement 1c, d*). For instance, if the decision boundary slowly drifts to the left side then rightward

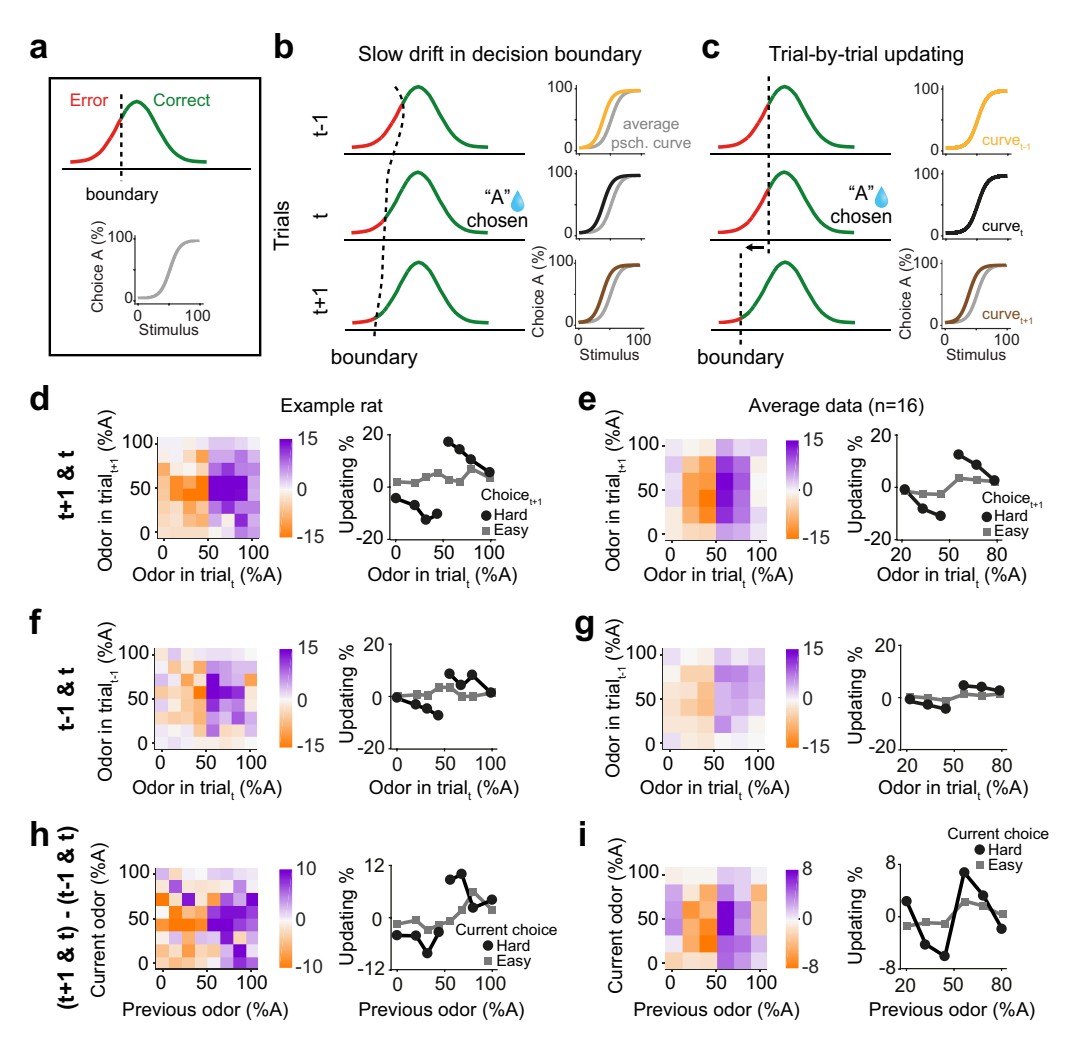

**Figure 2.** Choice updating is not due to slow and nonspecific drift in response bias. (**a**) Signal detection theory-inspired schematic of task performance. The psychometric curve illustrates the average choice behavior. (**b**) Slow non-specific drift in choice bias, visualized here as drift in the decision boundary, could lead to shift in psychometric curves which persisted for several trials and was not specific to stimulus and outcome of the previous trial. This global bias effect is cancelled when subtracting the psychometric curve of trial$_{t-1}$ (orange) from trial$_{t+1}$ (brown). (**c**) Trial-by-trial updating of decision boundary shifts psychometric curves depending on the outcome and perceptual difficulty of the preceding trial. Subtracting psychometric curves does not cancel this effect. (**d**) Choice bias of the example rat following a rewarded trial. (**e**) Similar to d but for population. (**f**) Choice bias of the example rat in one trial prior to current trial, reflecting global nonspecific bias visualized in b. (**g**) Similar to f but for population. (**h**) Subtracting choice bias in trial$_{t-1}$ from trial$_{t+1}$ reveals the trial-by-trial choice updating in the example rat. (**i**) Similar to h but for the population. See *Figure 2—figure supplement 1* for details of the normalization procedure.

The online version of this article includes the following figure supplement(s) for figure 2:

**Figure supplement 1.** Isolation and correction of slowly drifting non-specific choice bias.

choices will be more frequent and occur in succession, producing a shift in the psychometric curve (*Figure 2b*). Importantly, this effect is independent of decision outcomes and is observed in a sequence of trials, both before and after a rewarded trial (*Figure 2b*). An alternative, more intriguing, scenario for explaining our results is an active learning process that produces trial-by-trial adjustment of the decision boundary. If the decision boundary is adjusted in a trial-by-trial manner according to the outcome of the previous trial, psychometric shifts will be observed in the next trial contingent on the past reward, but absent in the preceding trial (*Figure 2c*). It is thus critical to remove slow fluctuations of side bias, before concluding that psychometric shifts are signatures of an active learning process.

We next asked to what extent the trial history effects we observed reflect correlations across consecutive trials due to slowly fluctuating bias over trials. To do so, we devised a model-independent analysis to identify and remove slow fluctuations of side bias (*Figure 2—figure supplement 1*). While it is possible to formulate model-based analyses to correct for slow biases, there are numerous possibilities that could produce similar fluctuations. Therefore, we sought a model-independent technique, reasoning that slow fluctuations are, by definition, slower than one trial, and hence should have largely similar impact across adjacent trials. Specifically, slow fluctuations will produce similar biases one trial before and after a given decision outcome. This assumption leads to a simple strategy to correct for possible slow drifts and isolate psychometric curve shifts due to active processes: subtracting the psychometric shifts between trial t and t-1 from that of trial t+1 and t removes the effect of slow response bias, that is slow boundary drift. Importantly, applying this normalization to the STD model with a drifting decision boundary removes the apparent but artefactual dependence of decisions on previous trials (*Figure 2—figure supplement 1d*). This subtraction technique thus provides an estimate of how the current trial influences choices in the next trial. Another intuition for this analysis is that future rewards cannot influence past choices, and therefore any systematic dependence of psychometric curves on next trials cannot reflect causal mechanisms and need to be adjusted for. We thus define 'choice updating' as a trial-by-trial bias beyond slowly fluctuating and non-specific side biases.

We found that the difficulty of previous choices had a strong effect on subsequent choices in rat olfactory discriminations even after correcting for slow fluctuations in the choice bias (*Figure 2d–i*). We computed the psychometric curves conditional on stimuli and outcomes, for both the next trial (*Figure 2d,e*) and also the previous trial (*Figure 2f,g*). Choice biases tended to be larger when considering the next trial compared to the previous trial. The difference between these provides an estimate of choice updating that is due to the most recent reward and without slow and non-outcome specific fluctuations in side bias (*Figure 2h,i*). The choice updating effect remained statistically significant even after this correction (p=0.003, rank sum test). These results rule out the possibility that psychometric shifts are only due to slow drift in side bias and indicate that reward received in the past trial influences subsequent perceptual decisions specifically if the sensory evidence in the previous trial was uncertain and difficult to judge.

## Belief-based reinforcement learning models account for choice updating

We next considered what types of reinforcement learning processes could account for the observed choice updating effects. Reinforcement learning models have been long used to study how choices are influenced by past decisions and rewards (*Daw and Doya, 2006*; *Sutton and Barto, 1998*). A key distinction between RL model variants is whether and how they treat ambiguous signals for state inference and prediction error computation.

We show that a reinforcement learning model with a belief-state representing ambiguous perceptual stimuli accounts for choice updating (*Figure 3a,b*). A reinforcement model for our behavioral task has to consider the inherent perceptual ambiguity in sensory decisions in addition to tracking reward outcomes. The normative way to cope with such ambiguity about state representations is to introduce a partially observable Markov decision process (POMDP) framework for the temporal difference RL (TDRL) algorithm (*Dayan and Daw, 2008*; *Lak et al., 2017*; *Lak et al., 2019*; *Rao, 2010*). POMDPs capture the intuitive notion that when perception is ambiguous, the model needs to estimate current perceptual experience as a 'belief state', which expresses state uncertainty as a probability distribution over states.

Previously, we showed that such a model is analogous to a TDRL model that uses statistical decision confidence, the conditional probability of getting reward given the choice, to scale prediction errors (*Lak et al., 2017*). We reasoned that this model could also account for how uncertainty of past perceptual decisions influences learning and updating of subsequent perceptual choice. In the model, the rewards received after difficult, low confidence, choices lead to large reward prediction errors and hence the strong updating of decision values in the next trial (*Figure 3a, b*). The belief state in the model reflects the subject's internal representations of a stimulus, which in the case of our 2AFC task is the probability that the stimulus belongs to the left or right category ($p_L$ and $p_R$). To estimate these probabilities, the model assumes that the internal estimate of the stimulus, $\hat{s}$, is normally distributed with constant variance around the true stimulus contrast: $p(\hat{s}|s) = N(\hat{s}; s, \sigma^2)$, where

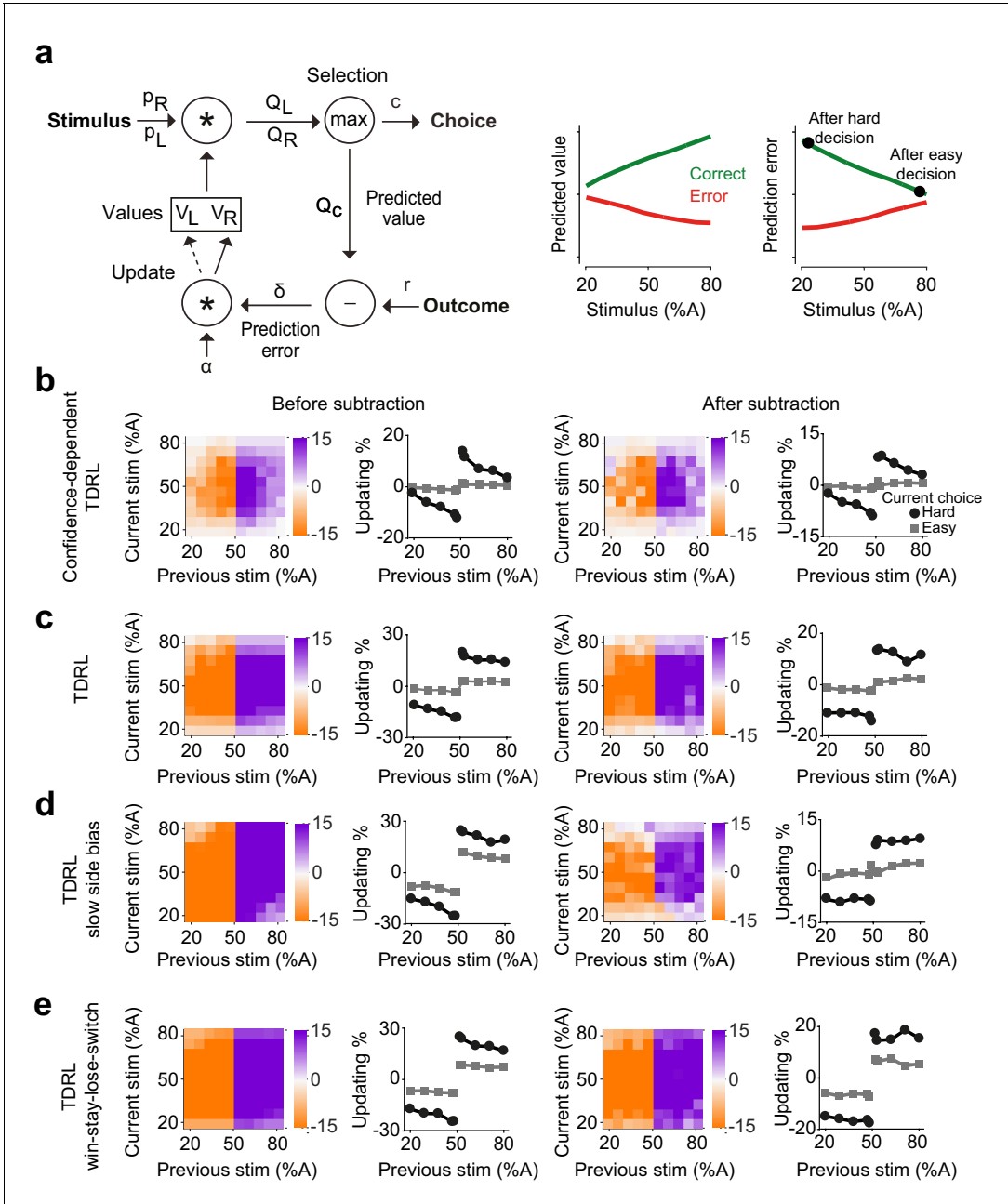

**Figure 3.** Belief-based reinforcement learning model accounts for choice updating. (a) Left: schematics of the temporal difference reinforcement learning (TDRL) model that includes belief state reflecting perceptual decision confidence. Right: predicted values and reward prediction errors of the model. After receiving a reward, reward prediction errors depend on the difficulty of the choice and are largest after a hard decision. Reward prediction errors of this model are sufficient to replicate our observed choice updating effect. (b) Choice updating of the model shown in a. This effect can be observed even after correcting for non-specific drifts in the choice bias (right panel). The model in all panels had $\sigma^2$=0.2 and $\alpha$=0.5. (c) A TDRL model which follows a Markov decision process (MDP) and that does not include decision confidence into prediction error computation produces choice updating that is largely independent of the difficulty of the previous decision. (d) A MDP TDRL model that includes slow non-specific drift in choice bias fails to produce true choice updating. The normalization removes the effect of drift in the choice bias, but leaves the difficulty-independent effect of past reward (e) A MDP TDRL model that includes win-stay-lose-switch strategy fails to produce true choice updating. For this simulation, win-stay-lose-switch strategy is applied to 10% of randomly-selected trials. See *Figure 3—figure supplement 1* and the Materials and methods for further details of the models.

The online version of this article includes the following figure supplement(s) for figure 3:

**Figure supplement 1.** Further characteristics of the confidence-dependent TDRL model and the MDP TDRL model.

$\hat{s}$ parameterizes a belief distribution over all possible values of $s$ that are consistent with the sensory evidence, given by Bayes rule:

$$p(s|\hat{s}) = \frac{p(\hat{s}|s).p(s)}{p(\hat{s})}$$

Assuming that the prior belief about $s$ is uniform, then the optimal belief will also be Gaussian, with the same variance as the sensory noise distribution, and mean given by $\hat{s}$: $p(s|\hat{s}) = N(s; \hat{s}, \sigma^2)$.

From this, the agent computes a belief, that is the probability that the stimulus was on the right side of the monitor, $p_R = p(s>0|\hat{s})$, according to:

$$p_R(\hat{s}) = \int_0^\infty p(s|\hat{s})ds$$

where $p_R$ represents the trial-by-trial probability of the stimulus being on the right side and $p_L = 1 - p_R$ similarly represents the probability of it being on the left. Multiplying these probabilities with the learned action values of left and right, $V_L$ and $V_R$, provides the expected values of left and right choices: $Q_L = p_L V_L$ and $Q_R = p_R V_R$. The higher of these two determines the choice $C$ (either $L$ or $R$), its associated confidence $p_C$, and its predicted value $Q_c = p_C V_C$. Note that although the choice computation is deterministic, the same stimulus can produce left or right choices caused by fluctuations in the percept due to randomized trial-to-trial variation around the stimulus identity (*Figure 3—figure supplement 1a*). Following the choice outcome, the model learns by updating the value of the chosen action by $V_C \leftarrow V_C + \alpha\delta$, where $\alpha$ is a learning rate, and $\delta = r - Q_C$ is the reward prediction error. Thus, in this model, prediction error computation has access to the belief state used for computing the choice, and hence reward prediction error is scaled by the confidence in obtaining the reward (*Lak et al., 2017*; *Lak et al., 2019*). The largest positive prediction error occurs when receiving a reward after a difficult, low confidence, choice while receiving a reward after an easy choice results in a small prediction error (*Figure 3a*, right). After training, the choices produced by this model exhibit confidence-guided updating (*Figure 3b*, left), similar to those we observed in choices of rats. Similar to the data, the choice updating in this model persisted after accounting for possible slow fluctuations in the choice bias (*Figure 3b*, right). Note that RL models can produce correlations in choices across trials due to the correlation of stored values across trials, and hence it is important to evaluate the size of the updating effect after the normalization. An additional prediction of our model is that updating effect should be slightly stronger when considering trials preceded by two (rather than one) rewarded difficult choices in the same direction (*Figure 3—figure supplement 1b*), which we also observed in rats' choices (*Figure 3—figure supplement 1b*).

We next considered whether classical TDRL models that follow a Markov decision process (MDP), could also produce confidence-guided updating. Such a model is largely similar to the model described above with one fundamental difference: the computation of prediction error does not have access to the belief state. In other words, prediction errors are computed by comparing the outcome with the average value of chosen action, without consideration of the belief in the accuracy of that action. In the model variant with two states (L and R), after learning, $V_L$ and $V_R$ reflect the average reward expectations for each choice and prediction errors are computed by comparing the outcome with this average expectation. Decisions made by this model show substantial effects of past reward (*Figure 3c*): after receiving a reward, the model has a tendency to make the same decision. However, compared to the belief-based model, this bias shows little dependence on the difficulty of the previous choice, and this dependence is absent after applying our normalization (*Figure 3c*, left). An extended version of this model that represents stimuli across multiple states also cannot reproduce confidence-dependent updating (see Materials and methods). Thus, MDP TDRL models (i.e. TDRL without a stimulus belief state) do not exhibit confidence-guided choice updating. We also considered whether a MDP TDRL model with slowly fluctuating response side bias could show trial-by-trial choice updating (*Figure 3d*). A modified MDP TDRL model that includes a slowly drifting side bias term showed a substantial effect of past reward on choices, which was mildly dependent on the difficulty of the previous sensory judgement (*Figure 3d*, left). This dependence, however, vanished after normalizing the choices to account for slowly drifting side bias (*Figure 3d*, right). This reveals that slowly fluctuating side bias in TDRL models without a belief state does not result in confidence-guided choice updating, despite apparent trial to trial fluctuations.

The results also further confirm the effectiveness of our normalization procedure (i.e. subtracting the bias in the previous trials from next trials) in isolating trial-by-trial confidence-guided choice updating. We also ruled out the possibility that choice updating reflected an elementary win-stay strategy (*Figure 3e*). The decisions of an MDP TDRL model that was modified to include win-stay strategy (to repeat the previously rewarded choice, with p=10% in this example) show strong dependence on past rewards. The normalization removes the dependence of updating size on choice difficulty that is due to a correlation across choices. However, it does not remove the signatures of win-stay / loose-switch behavior (*Figure 3e*). Thus, in this model variant the effect of past rewards is independent of the difficulty of the previous choice, differing from the rat data in which the size of the bias induced by previous reward is proportional to the difficulty of the previous decision.

The predictions of the confidence-dependent TDRL and MDP TDRL models differ in two principal ways. First, the learned values of actions ($V_L$ and $V_R$) converge to different values over learning (*Figure 3—figure supplement 1c*). In the confidence-dependent TDRL, $V_L$ and $V_R$ both converge to just below the true size of reward. However, in the MDP TDRL model they converge to the average choice accuracy (average reward harvest), which is lower than the true reward size (*Figure 3—figure supplement 1c*). This difference emerges because in the belief-based model values are updated using prediction errors scaled by confidence. Confidence is relatively low in the error trials, leading to small adjustments (reductions) of values after those trials, and hence the convergence of values to just below the true reward size (i.e. reward value, when the reward is given). This difference is important for understanding updating in particular in the case of very easy correct trials. In the confidence-dependent model, large confidence associated with a correct easy choice together with higher values of stored values produce $Q$ values similar to the true reward size, and hence near-zero prediction errors when receiving the reward. However, in the MDP model, $Q$ values are compared with relatively low stored values (compared to the reward value), and hence reward prediction errors persist even for very easy correct choices. The second major difference between the two models is how choice difficulty determines reward prediction errors and hence updating. While in the confidence-dependent model prediction errors depend on the choice difficulty (*Figure 3a*; *Figure 3—figure supplement 1d*), in the MDP TDRL the prediction errors do not reflect choice difficulty (*Figure 3—figure supplement 1d*, see Materials and methods). The prediction errors of the belief-based model thus result in graded levels of updating in the subsequent trials depending on the difficulty of the previous choice.

## On-line learning in margin-based classifiers explains choice updating

We next considered whether another class of models based on classifiers could also explain choice updating. Perceptual decision-making processes produce discrete choices from ambiguous sensory evidence. This process can be modelled with classifiers that learn a boundary in the space of sensory evidence. For instance, in our olfactory decision task evidence is a two-dimensional space spanned by the odor components A and B (*Figure 4a*). The hyperplane separating choice options creates a decision boundary determining whether each odor mixture is classified as A or B, which can be learned over trials. To apply these ideas to our decision tasks, we examine a probabilistic interpretation of Support Vector Machine (SVM) classifiers, a powerful technique from machine learning (see Materials and methods). SVMs learn classification hyperplanes that produce decision boundaries that are maximally far away from any data sample. Applying this framework to our particular decision problem requires an on-line implementation of a probabilistic interpretation of SVMs (*Sollich, 2002*). In the SVM terminology, the distance of a sample *x* from the category boundary or hyperplane is called the 'margin' of the data point (orange arrow in *Figure 4a*). The size of the margin for a stimulus is proportional to the likelihood of that data point belonging to a class given the current classifier. After appropriate normalization, this model yields an estimate of the classification success for a given decision. Indeed, the Bayesian posterior of classification success is a normative definition of confidence (*Hangya et al., 2016*; *Pouget et al., 2016*).

Simulation of this on-line, Bayesian SVM produces choice updating similar to that from the RL model (*Figure 4b, c*). Thus, statistical classifiers, in which the decision boundary is continuously updated in proportion to the estimated classification success (i.e. decision confidence), provides another account of choice updating. The core computational feature common to reinforcement learning and classifier models is that statistical decision confidence contributes to the trial-by-trial adjustment of choice bias.

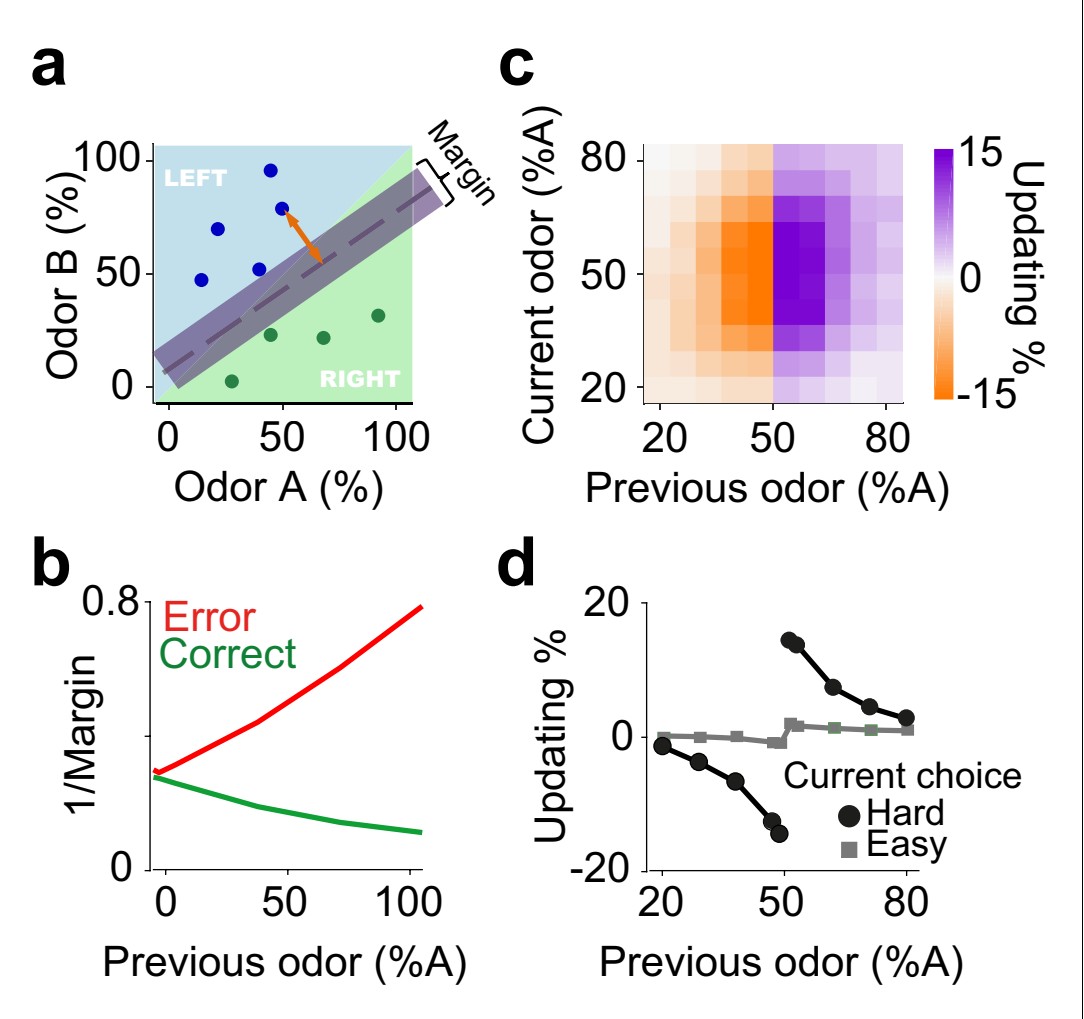

**Figure 4.** An on-line statistical classifier accounts for choice updating. (a) Schematic of a classifier using Support Vector Machine for learning to categorize odor samples. The dashed line shows one possible hyperplane for classification and shaded area around the dashed line indicates the margin. Orange arrow indicates the distance between one data point and the classification hyperplane, that is the margin for that data point, given the hyperplane. Each circle is one odor sample in one trial. (b) Average estimates of the margins of the classifier. (c) The size of shift in the classification as a function of previous and current stimulus. (d) Choice updating as a function previous odor separated for current easy and hard choices.

In the following sections, we examine whether confidence-guided choice updating is observed in sensory modalities other than olfaction, and in species other than rats.

## Confidence-guided choice updating in rat auditory decisions

We next investigated decisions of rats performing a two-alternative auditory decision task (*Figure 5*). Rats were trained to report which of two auditory click trains delivered binaurally had a greater number of clicks (*Sanders et al., 2016*; *Sanders and Kepecs, 2012*; *Brunton et al., 2013*; *Figure 5a*). The click trains were presented for 250 ms and generated using a Poisson process with the sum of the two rates held constant across trials. To control decision difficulty, the ratio of click rates for each side was randomly varied from trial to trial. The strength of evidence on a given trial was computed based on the number of clicks, that is the difference in the number of clicks between left and right divided by the total number of clicks. Rats showed steep psychometric curves (slope: 0.37±0.03, mean± S.D) with minor overall bias (-1±4.9% sound A) and near-zero lapse for easy stimuli

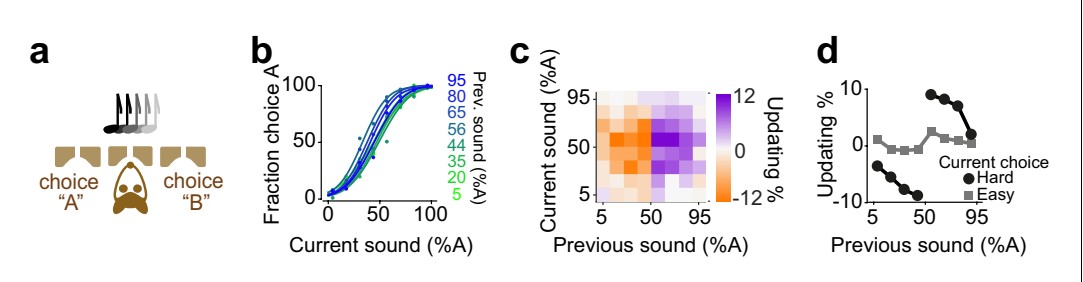

**Figure 5.** Rats update their trial-by-trial auditory choices in a confidence-dependent fashion. (**a**) Schematic of a 2AFC auditory decision-making task for rat. (**b**) Performance of an example rat computed separately based on the quality of auditory stimulus (shown as colors from blue to green) in the previously rewarded trial. (**c**) Choice updating as a function of sensory evidence in the previous and current trial in the population of rats (n = 5). (**d**) Choice updating as a function of previous stimulus separated for current easy (square) and difficult (circle) trials, averaged across rats.

(0.05±0.09%). We found that, similar to the olfactory task, difficult choices were biased in proportion to the difficulty of previous choice (*Figure 5b-d*, p=0.007, rank sum test).

## Confidence-guided choice updating in mouse auditory decisions

We next considered decisions of mice performing an auditory decisions task (*Figure 6*). Mice were trained on a two-alternative auditory tone decision task. The auditory stimuli in different trials were presented as percentages of a high and low frequency complex tone, morph A and morph B (*Figure 6a*). To manipulate perceptual uncertainty, we varied the amplitude ratio of the two spectral peaks in a trial-by-trial manner. Mice showed steep psychometric curves (slope: 0.25±0.06, mean ± S.D) with minor overall bias (-1±6.9% sound A) and negligible lapse for easy stimuli (3±2%). The choices showed significant dependence on the difficulty of previous choice (*Figure 6b-d*, p=0.01, rank sum test).

## Confidence-guided choice updating in mouse visual decisions

We next considered mice trained to perform visual decisions (*Figure 7*; *Burgess et al., 2017*). Head-fixed mice were trained to report the position of a grating on the monitor by turning a steering wheel placed under their front paws (*Figure 7a*). If the mouse turned the wheel such that the stimulus reached the center of the screen, the animal received water. If instead the mouse moved the stimulus by the same distance in the opposite direction, this incorrect decision was penalized with a timeout of 2 s (*Burgess et al., 2017*). We varied task difficulty by varying the contrast of the stimulus in different trials. Mice showed steep psychometric curves (slope: 0.260±11, mean ± S.D) with minor overall bias (1±9.1% stimulus contrast) and negligible lapse for easy stimuli (5±4.9%). The decisions showed significant dependence on the difficulty of previous choice (*Figure 7b-d*, p=0.001, rank sum

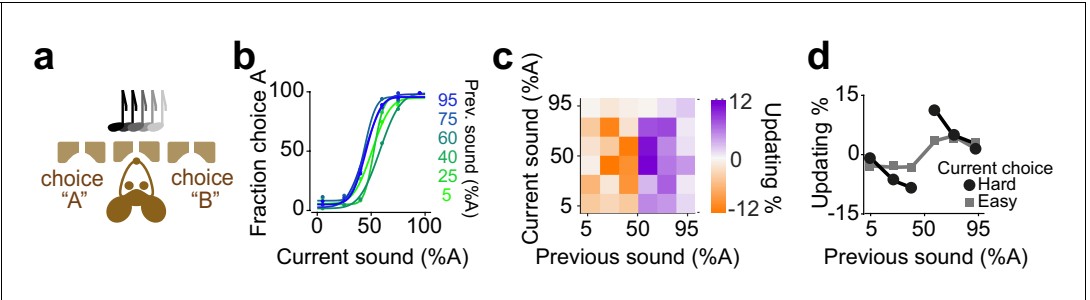

**Figure 6.** Mice update their trial-by-trial auditory choices in a confidence-dependent fashion. (**a**) Schematic of a 2AFC auditory decision making task for mice. (**b**) Performance of an example mouse computed separately based on the quality of auditory stimulus (shown as colors from blue to green) in the previously rewarded trial. (**c**) Choice updating as a function of sensory evidence in the previous and current trial in the population of mice (n = 6). (**d**) Choice updating as a function of previous stimulus separated for current easy (square) and difficult (circle) trials, averaged across mice.

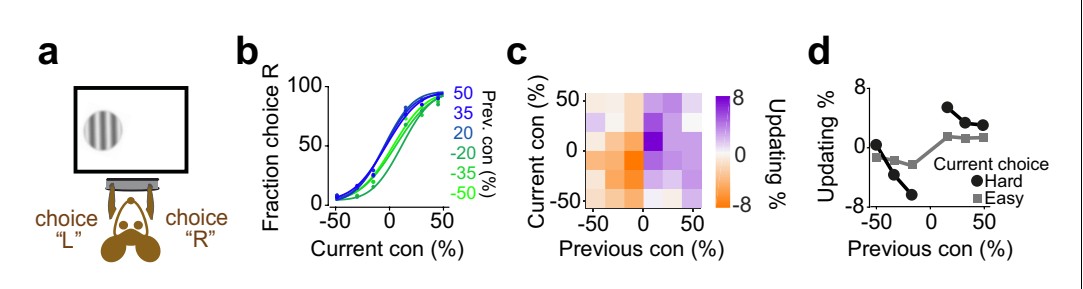

**Figure 7.** Mice update their trial-by-trial visual choices in a confidence-dependent fashion. (a) Schematic of a 2AFC visual decision making task for mice. (b) Performance of an example mouse computed separately based on the quality of visual stimulus (shown as colors from blue to green) in the previously rewarded trial. (c) Choice updating as a function of sensory evidence, that is the contrast of stimulus, in the previous and current trial in the population of mice (n = 12). (d) Choice updating as a function of previous stimulus separated for current easy (square) and difficult (circle) trials, averaged across mice.

test), consistent with similar results in a different version of this task that also included manipulation of reward size (*Lak et al., 2019*).

## Confidence-guided choice updating in human visual decisions

Next, we asked whether confidence-guided choice strategy was specific to rodents or could also be observed in humans (*Urai et al., 2017*). Human observers performing a visual decision task updated their choices in a confidence-dependent manner (*Figure 8*). Observers performed a two-interval forced choice (2IFC) motion coherence discrimination task (*Figure 8a*). They judged the difference in motion coherence between two successively presented random dot kinematograms: a constant reference stimulus (70% motion coherence) and a test stimulus with varying motion coherence in different trials (*Urai et al., 2017*). Observers performed the task well (slope: 0.2±0.01, bias: 0±3% stimulus coherence, lapse: 1±2%), and their choices showed significant dependence on the difficulty of previous choice (*Figure 8b-d*, P<0.05, rank sum test).

## Confidence-guided choice updating transfers across sensory modalities

We found that rats exhibit confidence-dependent choice updating even if the sensory modality of current decision differed from the modality of the previous decision (*Figure 9*). We trained rats in a dual sensory modality 2AFC task with randomly interleaved trials of auditory and olfactory decisions (*Figure 9a-b*). Rats performed the task well (slope: 0.35±0.03, bias: -2±6%, lapse: 1±0.4%) and their choices showed significant dependence on the difficulty of the past choice, and this dependence transferred across the sensory modalities. Rats updated their olfactory decisions after difficult auditory decisions (*Figure 9c-e* p=0.008, rank sum test), and similarly, they updated their auditory

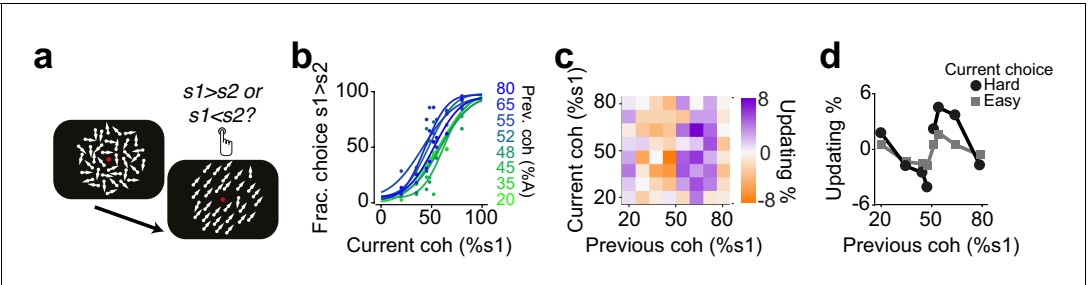

**Figure 8.** Humans update their trial-by-trial visual choices in a confidence-dependent fashion. (a) Schematic of a 2IFC visual decision making task in human subjects. (b) Performance of an example subject computed separately based on the quality of visual stimulus (shown as colors from blue to green) in the previously rewarded trial. (c) Choice updating as a function of sensory evidence, that is the difference in coherence of moving dots between two intervals, in the previous and current trial, averaged across subjects (n = 23). (d) Choice updating as a function of previous stimulus strength, separated for current easy (square) and difficult (circle) trials, averaged across subjects.

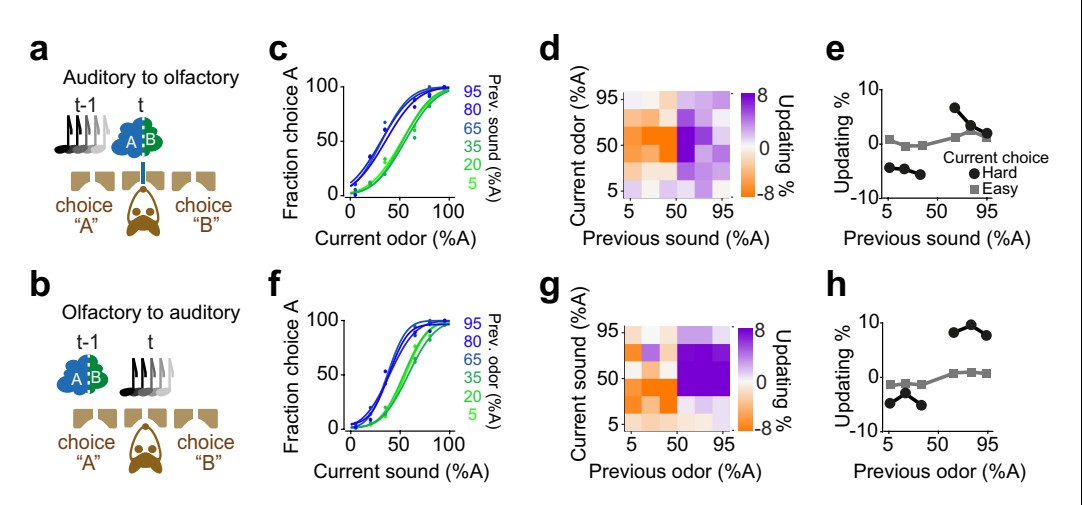

**Figure 9.** Confidence-dependent choice updating transfers across sensory modalities. (a-b) Schematic of a 2AFC task in which rats performed either an olfactory (a) or auditory (b) decisions in randomly interleaved trials. (c) Performance of an example rat computed for olfactory trials separately based on the quality of auditory stimulus (shown as colors from blue to green) in the previously rewarded trial. (d) Choice updating as a function of sensory evidence (auditory stimulus) in the previous trial and odor mixture in the current trial, averaged across subjects (n = 6). (e) Choice updating as a function of previous auditory stimulus separated for current odor-guided easy (square) and difficult (circle) trials, averaged across subjects. (f-h) Similar to c-e but for trials in which the current stimulus has been auditory and the previous trial has been based on olfactory stimulus.

The online version of this article includes the following figure supplement(s) for figure 9:

**Figure supplement 1.** Choice-updating in rats performing a task in which the modality of sensory stimulus in different trials is either auditory or olfactory.

choices after difficult olfactory decisions (*Figure 9f, h*, p=0.01, rank sum test). The updating occurred also across trials in which the modality of the choice did not change (i.e. in consecutive auditory choices and consecutive olfactory choices). Choice updating appeared largest in two consecutive olfactory decisions, yet updating was present in consecutive auditory choices as well as in consecutive trials with a modality switch (*Figure 9—figure supplement 1*). These results show that the updating effect depends on the choice outcome, rather than the identity, that is modality, of the sensory stimulus. This observation further indicates that choice updating mainly occurs in the space of action values, similar to our RL model.

## Diversity of confidence-guided choice updating across individuals

Having observed confidence-guided choice updating across various data sets, we next quantified this behavioral effect in each individual, and observed a weak but negative correlation between the strength of choice updating and psychometric lapse rate (*Figure 10*). To quantify the choice updating effect for each individual, we performed linear regressions on the updating data (*Figure 10a* inset) and computed the updating index as the difference in the slope of the fits for current easy and current difficult trials. A large fraction of individuals in each data set showed substantial choice updating consistent with the predictions of the model (positive numbers in *Figure 10a*). Nevertheless, we also observed individuals with negligible updating, and even in rare instances a choice bias in the direction opposite to the model's prediction (negative numbers in *Figure 10a*). Quantifying individual behavior enabled us to ask whether this observed diversity could be explained by variations in the quality of perceptual processing (psychometric slope and lapse rate; *Figure 10b,c*). The slope and updating did not exhibit a significant relationship (p=0.21), but the lapse rate and updating showed a weak significant negative correlation (p=0.03). In other words, choice updating was strongest among individuals with lower lapse rate (*Figure 10c*). The results suggest that choice updating of the form we observed, is strongest when subjects are well trained in the perceptual task, with stable psychometric slope and minimal lapse rate.

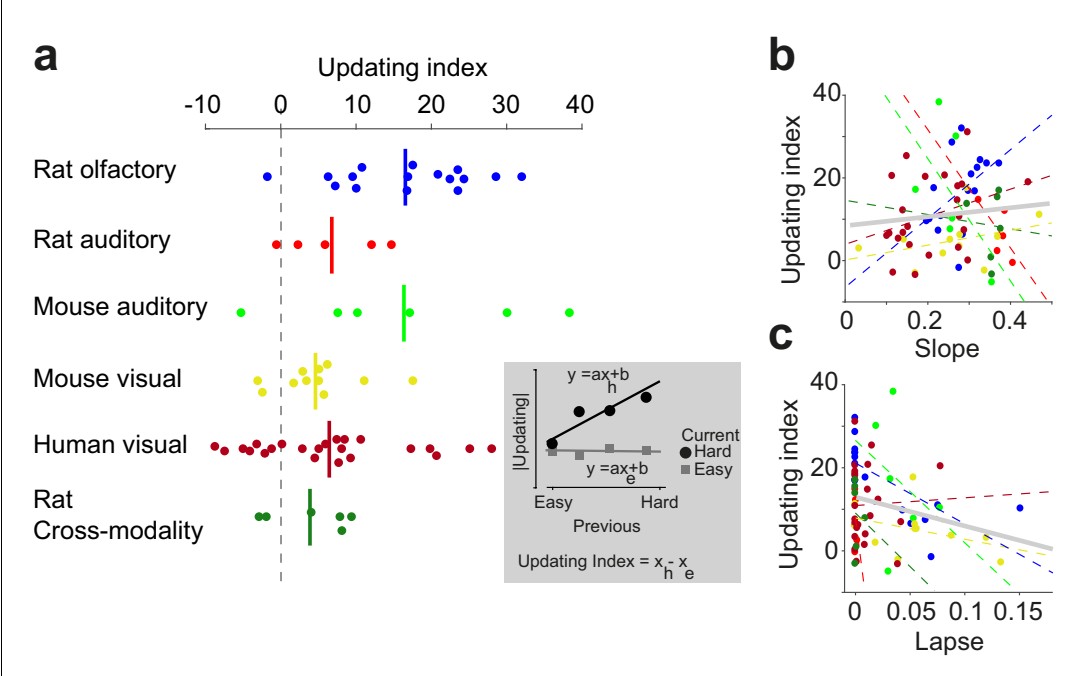

**Figure 10.** Confidence-guided choice updating is strongest in individuals with well-defined psychometric behavior. (**a**) The strength of choice updating among individuals. The vertical lines show the mean. Inset: schematics illustrates the calculation of updating index. The index is defined as the difference in the slope of lines fitted to the data. (**b**) Scatter plot of choice updating as a function of the slope of psychometric curve. Each circle is one individual. Dashed lines illustrate a linear fit on each data set, and the gray solid line shows a linear fit on all subjects. (**c**) Scatter plot of choice updating as a function of the lapse rate of the fitted psychometric curve.

## Different strategies for choice updating after error trials due to different noise sources

Lastly, we asked whether subjects show choice updating following error trials. We observed substantial diversity in choice bias after error trials, both across individuals and data sets. For example, two rats that performed olfactory decisions with similar choice updating after correct choices (*Figure 11a,b* top panel), showed divergent patterns of updating after incorrect choices (*Figure 11a,b* bottom panel). Interestingly, reinforcement learning models with different parameter settings also produced diverse choice bias patterns after error trials, depending on the dominant source of noise, that is whether errors were produced by sensory noise (external) or due to value fluctuations (internal). When sensory noise is high and leads to errors, these errors cannot be systematically corrected hence there is little or no net updating effect (*Figure 11c*, bottom panel). On the other hand, when the internal noise is high, such as high learning rate producing over-correction, systematic post-error updating is observed (*Figure 11d*, bottom panel). Although we found individual subjects matching these specific patterns of post-error updating (*Figure 11a,b*), the diversity across individuals and populations, as well as the low number of error trials precludes a more in-depth analysis. Note that for post-correct updating our RL model makes qualitatively similar predictions about choice updating, independent of whether the dominant source of noise is external or internal (*Figure 11c,d* top panels).

## Discussion

Our central observation is that even well-trained and well-performed perceptual decisions can be informed by past sensory evidence and outcomes. When the sensory evidence is strong, it determines choices, as expected, and past outcomes have little influence. When sensory evidence is weak, however, choices are influenced by trial history. We show that this effect of past rewards can depend on the difficulty of past sensory decisions: subjects repeat the previously rewarded choice

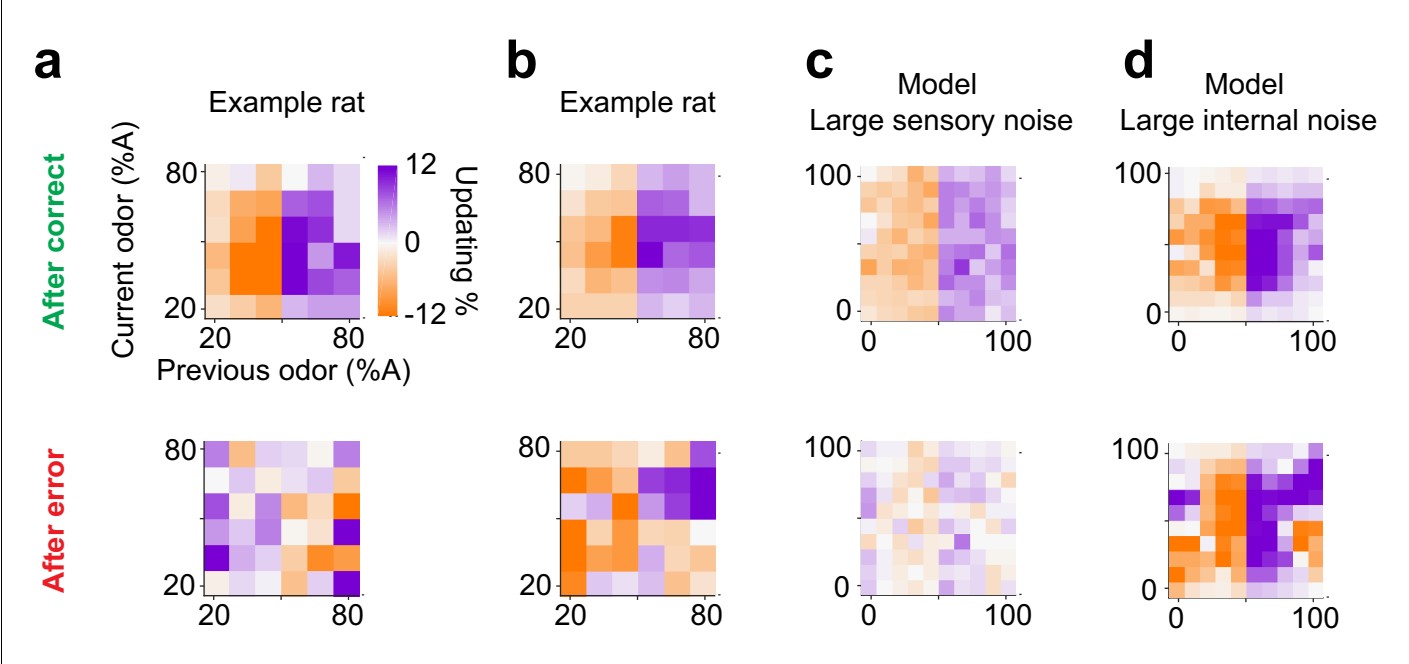

**Figure 11.** Diverse learning effects after error trials. (**a**) Choice updating after correct trials (top) and after error trials (bottom) in one example rat. (**b**) Similar to a for another example rat. (**c**) Choice updating of the TDRL model ran with large sensory noise ($\sigma^2 = 0.5$). This model exhibit choice updating qualitatively similar to the rat shown in a. (**d**) Choice updating in the TDRL model with large internal noise ($\alpha = 0.8$). This model run exhibits choice updating similar to the rat shown in b.

mainly when past sensory decision was difficult. This confidence-guided choice updating occurs in a trial-by-trial manner, and it is not due to slow drifts in choice strategy. We demonstrate that these history-dependent choice biases can be explained with reinforcement learning models that consider sensory ambiguity as a belief state computation, and hence produce confidence-scaled reward prediction error signals. We illustrate that this form of choice updating is a robust and widespread behavioral phenomenon observed across various perceptual decision-making paradigms in mice, rats and humans. Notably we found evidence for the same reinforcement learning process across data sets despite substantial variations in experimental setups and other conditions across the experiments examined.

The influence of past trials on perceptual decisions does not necessarily reflect active, trial-by-trial learning. In fact, our simulations illustrate that slow and non-specific drifts in the decision boundary result in correlations between consecutive choices, which can produce psychometric shifts similar to choice bias updating (*Figures 2* and *3*). To correct for this and isolate reinforcement learning-based choice updating, we used a simple procedure to compute choice updating with respect to slow fluctuations in the choice bias. We show the trial history effects persisting after this correction reflect confidence-guided reinforcement learning processes. These analyses also indicate that trial history effects and serial choice biases should be considered with care because correlations across choices at various time scales can produce apparent updating of perceptual choices. A similar confound has been previously reported in post-error slowing analysis, and similar normalization procedures were used to correct for it (*Dutilh et al., 2012*; *Purcell and Kiani, 2016*).

Confidence in the correctness of a choice determines the degree to which the reward can be expected. Hence, decision confidence informs how much the decision maker should learn from the decision outcome, as suggested by RL models that incorporate belief states representing confidence (*Lak et al., 2017*; *Lak et al., 2019*). Rewards received after decisions with high confidence are expected and hence there is not much to learn from them. In contrast, rewards received after decisions with lower confidence are relatively unexpected and could provide an opportunity to learn (*Figure 3*). Our results reveal that rodents and humans exhibit this form of learning (*Figure 10*).

These results were robust across various datasets, enabling us to isolate an elementary cognitive computation driving choice biases.

Trial-by-trial transfer of choice updating across sensory modalities provides some evidence that this form of learning is driven by comparing the decision outcome with confidence-dependent expectation and performing updating in the space of action values. However, updating across trials with different modalities was relatively weaker compared to trials within the same sensory modality. This later observation might point to the fact that in trial-by-trial learning animals follow a mixture of two strategies: one which updates values in the space of actions, and one that keeps track of stimulus identity and statistics across trials for such learning. The trade-off between these model-free and model-based trial-by-trial learning during perceptual decisions remains to be explored in future studies.

What are the neuronal substrates of confidence-dependent choice updating? Several lines of evidence indicate that the dopaminergic system is centrally involved in this phenomenon. First, dopamine neuron responses during perceptual decisions quantitatively match confidence-dependent prediction errors in both monkeys and mice (*Lak et al., 2017*; *Lak et al., 2019*). Second, dopamine responses predict the magnitude of the psychometric choice bias in the subsequent trial (*Lak et al., 2019*). Third, optogenetic manipulation of dopamine neurons biases psychometric curves in a trial-by-trial fashion (*Lak et al., 2019*). In addition, different frontal cortical regions, medial prefrontal cortex (*Lak et al., 2019*) and orbitofrontal cortex (*Hirokawa et al., 2019*) are also likely to contribute to confidence-guided choice updating strategies.

## Rewards induce choices bias in perceptual decisions

There is mounting evidence that in perceptual decision making tasks, even though reward is only contingent on accurate judgment about the current sensory stimulus, choices can be influenced by previous trials across species (*Abrahamyan et al., 2016*; *Akaishi et al., 2014*; *Akrami et al., 2018*; *Braun et al., 2018*; *Busse et al., 2011*; *Cho et al., 2002*; *Fan et al., 2018*; *Fischer and Whitney, 2014*; *Fritsche et al., 2017*; *Fründ et al., 2014*; *Gold et al., 2008*; *Hwang et al., 2017*; *Lueckmann et al., 2018*; *Luu and Stocker, 2018*; *Marcos et al., 2013*; *Tsunada et al., 2019*; *Urai et al., 2017*). Several such studies have shown that subjects might repeat the previously rewarded choice or avoid it after an unsuccessful trial, suggesting that basic forms of reward-based learning are at work even at asymptotic, steady-state perceptual performance. Given that these trial-history effects diminish the overall reward return, and are hence suboptimal, the question is why they persist even after subjects are well trained in the task? Our results showed a similar phenomenon across various data sets and species that produced choice bias in perceptual decisions after rewarded decisions that were difficult, consistent with recent reports (*Mendonça et al., 2018*; *Lak et al., 2019*). We show that these behavioral effects are normatively expected from various models that consider the uncertainty of stimulus states inherent in perceptual decisions. It is worth noting that the confidence-gauged learning described here requires observing the trial feedback, and it might thus differ from sequential choice effects in the absence of trial feedback (*Braun et al., 2018*; *Glaze et al., 2015*). Moreover, confidence-dependent learning differs from trial history-effects for highly discriminable stimuli, that is "priming of popout" effect (*Maljkovic and Nakayama, 1994*). These observations suggest that there are various types of selection-history mechanism operating in the brain with distinct constraints and properties.

## Computational mechanisms of confidence-driven choices bias

What classes of models can account for our behavioral observations? It is clear that a purely sensory-based model or a purely reward-based model cannot account for our data. One approach is to start with a reinforcement learning model and add a belief state to account for stimulus-induced uncertainty (*Lak et al., 2017*; *Lak et al., 2019*). We show that this class of models provides teaching signal reflecting past confidence and accounts for the observed choice bias strategy. Alternatively, we also considered a statistical classifier model that on-line adjusts the decision boundary in proportion to estimated classification success (*Sollich, 2002*). We show that this Bayesian on-line support vector machine also accounts for the observed choice bias strategy. Similarly, Bayesian learning in drift-diffusion models of decision making also makes similar predictions about confidence-dependent choice biases (*Drugowitsch et al., 2019*). Thus, either a sensory-based classification model modified to

produce statistically optimal adjustments based on on-line feedback, or a reward-based model modified to account for the ambiguity in stimulus states produce broadly similar confidence-dependent choice biases. These models share one main computation: they adjust the degree of learning based on the statistical confidence in the accuracy of previous decisions. Therefore, the key features of our data can be accounted for by either reinforcement learning mechanisms or on-line statistical classifiers.

These model classes can be distinguished chiefly on the basis of their respective decision variables: RL updates value, while classifiers update boundaries in sensory coordinates. Using a mixed sensory modality decision task it is possible to test whether updating is based on action values or stimulus variables. Actions value updating, predicted by RL models, leads to the transfer of choice bias across decisions with mixed sensory modalities. Category boundary updating in sensory coordinates, predicted by classifier models, leads to updating solely across the same sensory modality. We found that choice bias transferred across sensory modalities, suggesting that updating occurs in the space of action values. However, across-modality choice bias updating was weaker than within-modality updating, pointing to the possibility that animals update choices both based on stimulus statistics and action values.

## All correct trials are alike; each incorrect trial is incorrect in its own way

When learning from outcomes, it is natural to consider not only correct choices but also what happens after incorrect choices. Surprisingly, we found that post-error behavioral effects were highly variable across subjects and datasets, unlike the post-correct choice updating we observed.

To paraphrase Leo Tolstoy's famous opening sentence of the novel Anna Karenina: all correct trials are alike; each incorrect trial is incorrect in its own way. Correct perceptual performance requires appropriate processing and evaluation of the stimulus. In contrast there are many processes that can lead to incorrect performance without consideration of the stimulus, from inattention to lack of motivation to exploration. Indeed, in our behavioral data, the post-error behavioral effects were diverse, usually even within the same dataset.

We examined the RL model to gain insights into the possible origins of this post-error diversity. Note that the model's qualitative predictions about post-correct trials are largely independent of model parameters. In contrast, the predictions of the model for post-error trials depend on parameter settings, in part based on the balance in the sources of decision noise. When stimulus-noise was dominant there was little post-error updating. However, when the model's internal noise was high (e.g. large learning rate), it exhibited post-error updating. We found that individual subjects match these specific patterns but in fact the diversity in the data was greater, as expected, since the modeling framework does not take into account many relevant sources of decision noise, such as attentional lapses, lack of motivation or exploration, a multitude of processes that can all lead to errors.

Identifying the origin of decision noise is critical for the appropriate interpretation of any psychometric decision task. In the signal detection theory-based psychometric framework, negligible lapse rates and asymptotic psychometric slopes point to the interpretation that fluctuations in decisions are solely due to noise in perceptual processing. However, the ongoing learning mechanisms described here contribute to apparent fluctuations in the decisions, indicating that the contribution of sensory processing to decision noise may be often lower than previously thought (*Zariwala et al., 2013*). This ongoing learning might eventually go away after extended behavioral training yet we still observed signatures of this learning process in many subjects after 3–4 months of almost daily training.

## Consilience of perceptual and reward-guided decisions

Choice biases in perceptual decisions are typically considered maladaptive and suboptimal because in laboratory experiments trials are often designed to be independent from each other and isolate the perceptual process under study (*Britten et al., 1992*; *Hernández et al., 1997*). Indeed, perceptual choice biases that are entirely stimulus-independent are suboptimal. However, the confidence-guided choice bias we examined could point to an underlying optimal choice strategy from various perspectives. First, this strategy is optimal when considering that the world can change, and hence the precise decision category boundary may not be stable over trials. In this situation, the outcome of decisions closer to the category boundary provide the most informative feedback as to where the

category boundary should be set. Second, confidence-guided choice updating is optimal when considering that in natural environments external events can be temporally correlated. In such environments, when the evidence in favor of choice options is limited and hence there is uncertainty in decisions, it is beneficial to consider prior beliefs and the temporal statistics of events and adjust choices accordingly (*Yu and Cohen, 2008*). Thus, despite being suboptimal from the experimenter's perspective, confidence-guided learning can be optimal in dynamic, real world situations, revealing that perceptual and reinforcement learning processes jointly contribute to many previously studied decision paradigms.

## Materials and methods

### Data analysis and psychometric fitting

For the psychometric analysis, we calculated the percentage of choice as a function of sensory stimuli. We fitted these data with the psychometric function $\psi(s) = \lambda + (1 - 2\lambda)F(s; \mu, \sigma)$ where $F(x)$ is a cumulative Gaussian. The parameter μ represents the mean of the Gaussian and define the side bias. The parameter $\sigma$ determines the slope of the fitted curve. The parameter $\lambda$ represents the lapse rate of the curve. We fitted this function via maximum likelihood estimation (*Wichmann and Hill, 2001*).

We used the psychometric fits to evaluate whether the performance was substantially differed across days of testing in each subject. In addition, we computed conditional psychometric curves by computing the curves from a subset of trials, that is those that were preceded by specific stimulus level, action direction and outcome in the previous trial. We used the same procedure as above for fitting these conditional curves.

Subtracting the average performance (for each level of stimulus) from performance in the conditional curves for the same level of stimulus provided an estimate of choice updating, that is the level of side bias for each stimulus (*Figure 2*). The size of these side biases was plotted in the heatmaps for each dataset.

To isolate trial-by-trial updating independent of possible slow fluctuation in the choice bias, we estimated the slow side bias and subtracted it from the updating heatmaps (*Figure 2*, *Figure 2—figure supplement 1e*, *Figure 3*). This procedure involved computing conditional psychometric curves after a subset of trials (with specific stimulus, action and outcome) as well as computing the conditional curves prior to these trials, plotting heatmaps for both these sets of curves, and subtracting the later heatmap from the former heatmap (*Figure 2—figure supplement 1e*, *Figure 3*).

### Behavioral models

#### TDRL model with stimulus belief state

In order to examine the nature of choice updating during perceptual decision making, we adopted a reinforcement learning model which accommodate trial-by-trial estimates of perceptual uncertainty (*Lak et al., 2017*; *Lak et al., 2019*). In all our tasks, the subject can select one of two responses (often left vs right) to indicates its judgement about the stimulus (i.e. whether it belongs to category A or B, or left or right for simplicity). Knowing the state of the trial (left or right) is only partially observable, and it depends on the quality of sensory evidence.

In keeping with the standard psychophysical treatments of sensory noise, the model assumes that the internal estimate of the stimulus, $\hat{s}$, is normally distributed with constant variance around the true stimulus contrast: $p(\hat{s}|s) = N(\hat{s}; s, \sigma^2)$. In the Bayesian view, the observer's belief about the stimulus $s$ is not limited to a single estimated value $\hat{s}$. Instead, $\hat{s}$ parameterizes a belief distribution over all possible values of $s$ that are consistent with the sensory evidence. The optimal form for this belief distribution is given by Bayes rule:

$$p(s|\hat{s}) = \frac{p(\hat{s}|s).p(s)}{p(\hat{s})}$$

We assume that the prior belief about $s$ is uniform, which implies that this optimal belief will also be Gaussian, with the same variance as the sensory noise distribution, and mean given by $\hat{s}$:

$p(s|\hat{s}) = N(s; \hat{s}, \sigma^2)$. From this, the agent computes a belief, that is the probability that the stimulus was indeed on the right side of the monitor, $p_R = p(s>0|\hat{s})$, according to:

$$p_R(\hat{s}) = \int_0^{\infty} p(s|\hat{s})ds$$

where $p_R$ represents the trial-by-trial probability of the stimulus being on the right side and $p_L = 1 - p_R$ similarly represents the probability of it being on the left.

The expected values of the two choices L and R are computed as $Q_L = p_L V_L$ and $Q_R = p_R V_R$, where $V_L$ and $V_R$ represent the stored values of L and R actions. Over learning these values are converging to a quantity just below the true reward size available in correct choices (*Figure 3—figure supplement 1c*). To choose between the two options, we used an argmax rule which selects the action with higher expected value deterministically. Using other decision functions such as softmax did not substantially change our results (*Figure 3—figure supplement 1a*). The outcome of this is thus the choice $c$ (L or R), its associated confidence $p_C$, and its predicted value $Q_C$.

$$Q_C = \begin{cases} Q_L \, if \, choice = L \\ Q_R \, if \, choice = R \end{cases}$$

When the trial begins, the expected reward prior to any information about the stimulus is $V_{trialonset} = (V_L + V_R)/2$. Upon observing the stimulus and making a choice, the prediction error signal is: $Q_C - V_{trialonset}$. After receiving the reward, $r$, the reward prediction error is $\delta = r - Q_C$.

Given this prediction error, the value of the chosen action will be updated according to: $V_C \leftarrow V_C + \alpha.\delta$ where $\alpha$ is the learning rate. For simplicity, the model does not include temporal discounting. Parameter values used in *Figure 3* are: $\sigma^2$=0.2 and $\alpha$=0.5. Each agent received 500 trials per stimulus level (randomly presented to the model), and plots reflect averages across 1000 agents.

## TDRL models without stimulus belief state

The TDRL model without the belief state was largely similar to the model described above with one fundamental difference; computation of prediction error did not have access to the belief state used for choice computation. In other words, prediction errors are computed by comparing the outcome with the average value of chosen action, without consideration of the belief in the accuracy of that action.

Similar to the model above, the expected values of the two choices L and R are computed as $Q_L = p_L V_L$ and $Q_R = p_R V_R$, where $V_L$ and $V_R$ represent the stored values of L and R actions. Over learning these values are converging to the average value of reward received in the past trials, that is average performance in the task (*Figure 3—figure supplement 1c*). To choose between the two options, we used an argmax rule which selects the action with higher expected value deterministically. The outcome of this is thus the choice (L or R), its associated confidence $p_C$, and its predicted value $Q_C$.

$$Q_C = \begin{cases} Q_L \, if \, choice = L \\ Q_R \, if \, choice = R \end{cases}$$

After receiving the reward, $r$, the reward prediction error is $\delta = r - V_C$, where $c$ is choice, as before. Given this prediction error, the value of the chosen action will be updated according to:

$$V_C \leftarrow V_C + \alpha.\delta$$

Because $V_C$ does not reflect decision confidence, the prediction errors only reflect the presence or absence of reward, but are not modulated by the decision confidence (*Figure 3—figure supplement 1d*). Thus, they drive learning based on past outcome but not past decision confidence. Note that prior to normalization, there is an apparent small tendency for this model to exhibit updating that depends on the difficulty of the previous choice (*Figure 3c*, left). This reflect the correlation of stored values across trials and transient regimes in these stored values that make it more probable to achieve two consecutive correct same-side difficult choice. However, after the normalization, the

updating does not show any effect of previous difficulty, and merely reflects the presence or absence of reward in the previous trial (*Figure 3c*, right).

An extended version of this model is the one that includes multiple states: that is one state for storing the average value of each stimulus level. In this model, reward prediction error can be computed in two ways: with or without access to the inferred state. The first scenario makes prediction errors independent of past difficulty. The second scenario has access to the inferred state and compares reward with the value of that state to produce confidence-dependent prediction errors. However, since the updates only impact the current state there is no learning expected for different (nearby) stimuli.

## Modifications to include slowly drifting choice bias

We modified the TDRL model without the belief state to include slowly drifting side bias (Figure 3d). The bias over trials was defined as the moving average (across 50 trials) of $sin(t + A) + B$, where $A$ is the temporal noise over trials (drawn from a normal distribution, $N(0, 1)$), and $B$ is the amplitude noise (drawn from a normal distribution, $N(0, 1)$. On each trial, the bias (a negative or positive number), was added to $Q_L$, and choice was made by comparing $Q_L$ and $Q_R$, as described before. This induced drift causes a strong correlation across trials that influence choices. However, the normalization removes this correlation and the model shows an effect of past reward which is independent of the difficulty of the past stimulus.

## Modifications to include with win-stay-lose-switch strategy

We incorporated a simple probabilistic win-stay-lose-switch strategy into the TDRL model without the stimulus belief state. To do so, in 10% of randomly chosen trial, the choice was determined by the choice and outcome of the previous trial, rather than comparing $Q_L$ and $Q_R$. In these trials, the previous choice was repeated when the previous trial was rewarded (win-stay), or the alternative choice was reported if the previous trial was not rewarded (lose-switch).

## On-line Bayesian support vector machine (SVM)

The SVM algorithm finds the decision hyperplane that maximizes the margin between the data points belonging to two classes. The margin refers to the space around the classification boundary in which there is no data point, that is the largest minimal distance from any of the data points (*Figure 4a*). The data points falling on the margin are referred to as support vectors.

Following the conventional form of a linear SVM, stimuli are represented as $x_i$ each representing the two odor components A and B, and $y_i \in \{-1, 1\}$ corresponding to different class labels reflecting dominant odor A or B in the stimulus. The algorithm finds the hyperplane in this feature space that separate the stimuli. Assuming that the classification is determined by the following decision function $f(x) = sign(wT.\phi(x) + b)$, the classification hyperplane would be $w.x + b = 0$, where $w$ is a weight vector and $b$ is an offset. Thus, data point $x$ is assigned to the first class if $f(x) = sign(wT.\phi(x) + b)$ equals +1 or to the second class if $f(x)$ equals $-1$. In the simplest form (linear separability and hard margin), it is possible to select two parallel hyperplanes that separate the two classes of data, the distance between these hyperplanes is $2/\|w\|$, and hence to maximize this distance, $\|w\|$ should be minimized. In a more general form, the weights $w$ can be found by minimizing the following equation subject to the SVM constraints (i.e. no data point within the margin):

$$\frac{\lambda}{2}\|w\|_2^2 + \sum_i 1 - y_i w^T x_i$$

where $\lambda$ is the optimization hyperparameter that determines the trade-off between increasing the margin size and ensuring that the $x_i$ lie on the correct side of the margin. The equation above thus ensures a tradeoff between classification errors and the level of separability.

In the online active form, weights can be iteratively adjusted. In this form, is updated iteratively according to

$$w = \begin{cases} w + \alpha(-\lambda w + yx) \, if \, yw^T x < 1 \\ w - \alpha \lambda w \, if \, yw^T x >= 1 \end{cases}$$

where $\alpha$ is a learning rate. In Bayesian SVMs the size of the margin for one data point is proportional

to the likelihood of that point belonging to a class given the classifier. Thus, the posterior class probabilities, $P(y_i = +1|x_i, D)$, can be obtained by integrating over the posterior distribution of $w$ and $b$ after appropriate normalization. This quantity is proportional to the statistical decision confidence, that is an estimate of confidence about which odor mixture component predominates (*Figure 4b*). In simulation presented in *Figure 4*, we used $\alpha = 0.15$ as the learning rate for updating of weight vector and the $\sigma^2 = 0.25$ for underlying noise level of stimuli to be classified. Each agent received 500 trials per stimulus level (randomly presented to the model), and plots reflect averages across 1000 agents.

## Behavioral experiments

The experimental procedures were approved by Institutional committees at Cold Spring Harbor Laboratory (for experiments on rats), MIT and Harvard University (for mice auditory experiments) and were in accordance with National Institute of Health standards (project ID: 18-14-11-08-1). Experiments on mice visual decisions were approved by the home Office of the United Kingdom (license 70/8021). Experiments in humans were approved by the ethics committee at the University of Amsterdam (project ID: 2014-BC-3376).

## Rat olfactory experiment

The apparatus and task have been described previously (*Hirokawa et al., 2019*; *Kepecs et al., 2008*; *Uchida and Mainen, 2003*). The apparatus was controlled using PulsePal and Bpod (*Sanders and Kepecs, 2014*) or with the BControl system (https://brodylabwiki.princeton.edu/bcontrol/index.php?title=Main_Page). Rats self-initiated each experimental trial by introducing their snout into the central port where odor was delivered. After a variable delay, drawn from a uniform random distribution of 0.2–0.5 s, a binary mixture of two pure odorants, S(+)−2-octanol and R()−2-octanol, was delivered at one of six concentration ratios (80:20, 65:35, 55:45, 45:55, 35:65, 20:80) in pseudo-random order within a session. In some animals we also used ratios of 100:0 and 0:100. After a variable odor sampling time up to 0.7 s, rats responded by withdrawing from the central port, and moved to the left or right choice port. Choices were rewarded with 0.025 ml of water delivered from stainless tube inside of the choice port according to the dominant component of the mixture, that is, at the left port for mixtures A/B > 50/50 and at the right port for A/B < 50/50. The task training for these data and other dataset presented here focused on achieving high accuracy and did not specifically promote the confidence-dependent choice bias we have observed.

## Rat auditory experiment

Rats self-initiated each trial by entering the central stimulus port. After a random delay of 0.2–0.4 s, auditory stimuli were presented. Rats had to determine the side with the higher number of clicks in binaural streams of clicks (*Brunton et al., 2013*; *Sanders et al., 2016*; *Sanders and Kepecs, 2012*). Auditory stimuli were Poisson-distributed click trains played binaurally at the two speakers placed outside of the behavioral box for a fixed time of 0.25 s. For each rat, we chose a maximum click rate according to the performance of the animal, typically 50 clicks/s. This maximum click rate was fixed for each animal. For each trial, we randomly chose a delta click rate between left and right from a uniform distribution between 0 and maximum click rate. The sum of the left and right click rate was kept constant at maximum click rate. Rats indicated their choice by exiting the stimulus port and entering one of two choice ports (left or right) with a maximum response time of 3 s after leaving the stimulus port. Choices were rewarded according the higher number of clicks presented between the left and right click train. Exiting the stimulus port during the pre-stimulus delay or during the stimulus time (first 0.25 s) were followed by a white noise and a time out of 3–7 s.

## Rat randomly interleaved auditory-olfactory experiment

Rats self-initiated each trial by entering the central stimulus port. After a random delay of 0.2–0.4 s, either an olfactory or auditory stimulus was presented (randomly interleaved). For olfactory stimuli, rats had to determine the dominant odor of a mix of pure odorants +2-octanol and –2-octanol. Odor stimuli were delivered for at least 0.35 s or until the rat left the center port (max. 3 s). Odor mixtures were fixed at seven concentration ratios, which we adjusted to match the performance levels for each mixture ratio across animals, as described above in the rat olfactory experiment. One of the stimuli was a 50/50 ratio stimulus for which correct side is randomly assigned and those trials

were removed from the analysis. After a variable odor sampling time, rats exited the stimulus port, which terminated odor delivery, and indicated their choice by entering one of two choice ports (left or right) with a maximum response time of 3 s after leaving the stimulus port. Choices were rewarded according to the dominant odor component in the mixture. For auditory stimuli, rats had to determine the side with the higher number of clicks in binaural streams of clicks. Auditory stimuli were random Poisson-distributed click trains played binaurally at the two speakers placed outside of the behavioral box for a fixed time of 0.25 s, as described above for the rat auditory experiment. Exiting the stimulus port during the pre-stimulus delay or during the stimulus time (first 0.35 s for olfactory trials) were followed by a white noise and a time out of 3–7 s. Choices were rewarded according the higher number of clicks presented between the left and right click train. Reward timing was sampled from a truncated exponential distribution: minimum reward delay was 0.6 s, maximum delay 8 s and decay tine constant of 1.5 s.

## Mouse auditory experiment

Mice were water restricted for a week (administered 1.2 ml of water in a single session per day), handled for 2 days and then gradually shaped for 5–7 sessions to the contingencies of a two-alternative choice paradigm and subsequently trained to discriminate two complex stimuli before introducing 6 morphs of those stimuli. Trials were self-initiated upon the breaking of an infrared beam by a nose poke into the center port of three adjacent ports. Once mice remain in the center port over 0.2 s, mice were presented with one of two complex tones, following a 0.2–0.5 s delay (uniformly distributed). Auditory cues were presented until the mouse exited the center port. If a mouse entered the correct side port within 4 s, a 4 µl water drop was delivered from gravity-fed reservoirs regulated by solenoid valves. Trials in which mice did not remain in the center port long enough to elicit a cue were not considered valid trials and are not represented in our analyses. During the training period only, error trials were followed by a progressively increasing 3–10 s timeout in order to prevent rapid guessing. During initial training of the task, two complex tones were used for training. These are comprised of three tones centered on 3 kHz and three tones centered on 7.5 kHz, all components share base frequency of 1.5 kHz. These two training tones are described as 0 and 100 (%A) stimuli, respectively. In the perceptual decision-making task, each of the 36 complex tones varied in the balance of 6 components of Tones A and B. To vary discrimination difficulty, we varied the amplitude ratio of the two spectral peaks (3 kHz and 7.5 kHz). Morph tones comprised six sets of auditory stimuli described here as percentages of a high- and low-frequency complex tone, morph A and morph B, respectively. Each of six sets is comprised of six similar stimuli with percentages in terms of morph A of 5–10, 25–30, 35–40, 60–65, 70–75, 90–95. We thereby challenged mice with a variety of 36 stimuli, and were able to pool members of each stimulus set for the analysis. Stimuli were delivered through generic electromagnetic dynamic speakers located on each side of the behavior chamber.

## Mouse visual experiment

Mice were trained in a 2-alternative forced choice visual detection task (*Burgess et al., 2017*). After mouse kept the wheel still for at least 0.5 s, a sinusoidal grating stimulus of varying contrast appeared on either the left or right monitor, together with a brief tone (0.1 s, 12 kHz) indicating that the trial had started. The mouse could immediately report its decision by turning the wheel located underneath its forepaws. Wheel movements drove the stimulus on the monitor, and a reward was delivered if the stimulus reached the center of the middle monitor (a successful trial), but a 2 s white noise was played if the stimulus reached the center of the either left or right monitors (an error trial). The inter trial interval was set to 3 s. As previously reported, well-trained mice often reported their decisions using fast stereotypical wheel movements (*Burgess et al., 2017*). After 2–3 weeks of training, the task typically included 6 or 7 levels of contrast (three on the left, three on the right) which were presented in a random order across trials with equal probability.

## Human visual experiment

The experiments are described in detail in *Urai et al. (2017)*. Observers performed a two-interval forced choice motion coherence discrimination task at constant luminance. Specifically, observers judged the difference in motion coherence between two successively presented random dot kinematograms (RDKs): a constant reference stimulus (70% motion coherence) and a test stimulus (varying

motion coherence levels specified below). A beep indicated the onset of each (test and reference) stimulus. The intervals before, in between, and after (until the inter-trial interval) these two task-relevant stimuli had variable duration and contained randomly moving dots. After offset of the test stimulus, observers had 3 s to report their judgment (button press with left or right index finger, counterbalanced across observers). After a variable interval (1.5–2.5 s), a feedback tone was played. Dot motion was stopped 2–2.5 s after feedback, with stationary dots indicating the inter-trial interval, during which observers were allowed to blink their eyes. Observers self-initiated the next trial by button press. The difference between motion coherence of test and reference was taken from three sets: easy (2.5, 5, 10, 20, 30), medium (1.25, 2.5, 5, 10, 30) and hard (0.625, 1.25, 2.5, 5, 20). All observers started with the easy set and were switched to the medium set when their psychophysical thresholds (70% accuracy) dropped below 15%, and to the hard set when thresholds dropped below 10%, in a given session. Motion coherence differences were randomly shuffled within each block.

## Acknowledgements

This work was supported by the Wellcome Trust (grants 106101 and 213465 to AL). NU was funded by NIH R01 MH110404 and Mind Brain Behavior (MBB) Faculty Award. AK was funded by NIH R01MH097061 and R01DA038209. MC was funded by by the Wellcome Trust (205093). AEU was funded by a German Academic Exchange Service (DAAD) doctoral scholarship. THD was funded by German Research Foundation (DFG) grants: DO 1240/2–1 and DO 1240/3–1. ST and EH were supported by RIKEN-CBS, JPB Foundation, and HHMI.

## Additional information

### Competing interests

Naoshige Uchida: Reviewing editor, *eLife*. Tobias H Donner: Reviewing editor, *eLife*. The other authors declare that no competing interests exist.

### Funding

| Funder | Grant reference number | Author |
| --- | --- | --- |
| Wellcome | 106101 | Armin Lak |
| National Institutes of Health | R01MH110404 | Naoshige Uchida |
| Wellcome | 205093 | Matteo Carandini |
| Deutsche Forschungsgemeinschaft | DO 1240/2-1 | Tobias H Donner |
| RIKEN | | Emily Hueske<br>Susumu Tonegawa |
| JPB Foundation | | Emily Hueske<br>Susumu Tonegawa |
| Howard Hughes Medical Institute | | Emily Hueske<br>Susumu Tonegawa |
| German Academic Exchange Service | | Anne E Urai |
| National Institutes of Health | R01DA038209 | Adam Kepecs |
| Harvard University | Mind Brain Behavior (MBB) Faculty Award | Naoshige Uchida |
| Deutsche Forschungsgemeinschaft | DO 1240/3-1 | Tobias H Donner |
| Wellcome | 213465 | Armin Lak |
| National Institutes of Health | R01MH097061 | Adam Kepecs |

The funders had no role in study design, data collection and interpretation, or the decision to submit the work for publication.

### Author contributions
Armin Lak, Conceptualization, Data curation, Software, Formal analysis, Resources, Funding acquisition, Investigation, Visualization, Methodology; Emily Hueske, Junya Hirokawa, Paul Masset, Torben Ott, Anne E Urai, Data curation, Investigation, Methodology; Tobias H Donner, Matteo Carandini, Susumu Tonegawa, Resources, Supervision, Funding acquisition; Naoshige Uchida, Conceptualization, Resources, Supervision, Funding acquisition; Adam Kepecs, Conceptualization, Resources, Formal analysis, Supervision, Funding acquisition

### Author ORCIDs
Armin Lak https://orcid.org/0000-0003-1926-5458
Junya Hirokawa http://orcid.org/0000-0003-1238-5713
Paul Masset http://orcid.org/0000-0003-2001-7515
Anne E Urai http://orcid.org/0000-0001-5270-6513
Tobias H Donner http://orcid.org/0000-0002-7559-6019
Matteo Carandini http://orcid.org/0000-0003-4880-7682
Naoshige Uchida http://orcid.org/0000-0002-5755-9409

### Ethics
Human subjects: The ethics committee at the University of Amsterdam approved the study, and all observers gave their informed consent.project ID: 2014-BC-3376.
Animal experimentation: The experimental procedures were approved by Institutional committees at Cold Spring Harbor Laboratory (for experiments on rats), MIT and Harvard University (for mice auditory experiments) and were in accordance with National Institute of Health standards (project ID: 18-14-11-08-1). Experiments on mice visual decisions were approved by the home Office of the United Kingdom (license 70/8021). Experiments in humans were approved by the ethics committee at the University of Amsterdam (project ID: 2014-BC-3376).

### Decision letter and Author response
Decision letter https://doi.org/10.7554/eLife.49834.sa1
Author response https://doi.org/10.7554/eLife.49834.sa2

## Additional files

### Supplementary files
• Transparent reporting form

### Data availability
The human dataset used in this study is available at https://doi.org/10.6084/m9.figshare.4300043.

The following previously published dataset was used:

| Author(s) | Year | Dataset title | Dataset URL | Database and Identifier |
|---|---|---|---|---|
| Urai AE, Braun A, Donner THD | 2018 | Pupil-linked arousal is driven by decision uncertainty and alters serial choice bias | http://dx.doi.org/10.6084/m9.figshare.4300043 | Figshare, 10.6084/m9.Figshare.4300043 |

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
