## [Decision Letter]

**Acceptance summary:**

This study investigates how past choice history influences decisions that, in principle, should be guided just by currently available sensory information. It combines two classic approaches, perceptual decision making and reinforcement learning, to show that the success of prior choices and the quality of the perceptual information guiding them are automatically tracked, and that both factors are used to bias upcoming choices that are difficult, i.e., those for which the quality of the sensory information is poor and the subject is largely guessing. The effect is robust across sensory modalities, species, task details, and labs. It explains an important source of variance in behavior.

**Decision letter after peer review:**

Thank you for sending your article entitled "Confidence-guided updating of choice bias during perceptual decisions is a widespread behavioral phenomenon" for peer review at *eLife*. Your article has been evaluated by three peer reviewers, one of whom is a member of our Board of Reviewing Editors, and the evaluation has been overseen by Michael Frank as the Senior Editor.

All reviewers found the work interesting and appealing. However, some potentially

serious issues were identified that could undermine the main conclusions. The full set of recommendations is below. Please focus your revisions on points 1, 2, and 4, which are most critical.

Essential revisions:

1) The first major concern is in regard to a claim made throughout the paper, namely that how easy or hard the *current* stimulus is affects the strength of the modulation. This is first presented in Figures 1F and 1G, and then repeated in almost all the figures. However, the measure presented in figures of the style of Figure 1F and summarized in Figure 1G is the% change in responses (given a particular current and previous stimulus, and relative to the overall average for that stimulus). This measure suffers from a large problem that, unfortunately, appears to not be addressed (apologies if it was missed on reading): there is a ceiling effect, in the sense that when a choice is easy (near 100%) it cannot increase further, even if there were strong underlying plasticity. The ceiling applies preferentially to easy stimuli, and therefore will manifest as a difference between easy and hard stimuli. It is currently impossible to tell whether the data shown in support of the claim of "current stimulus strength modulates effect" is merely an artifact resulting from this ceiling issue, or an actual finding. It is very important that the authors clarify this issue.

2a) Another key claim is that only the model with a belief state could account for the results. But the data are not convincing. Let's start with a TD model, without learning of actions, merely learning the values *V_R_* and *V_L_*, and let's suppose the value of V_R_ is modified only after *R* choices (and similarly for *V_L_* and *L* choices). Under these conditions (and assuming *R-L* symmetry) *V_R_* and *V_L_* will converge to the overall% correct. This will be more than 50%; for a typical smooth psychometric curve it might be 75% or 80%. Now move to a TDRL model that also learns actions: because there are more errors for hard stimuli, the error signal (*V_C_ – r*) will, on average, be greater for hard stimuli than for easy stimuli. And therefore, in this situation, there will on average be greater plasticity after hard stimuli than after easy stimuli, entirely without a belief state. Could this not account for the authors' experimental data?

The situation and conclusion sketched here could be wrong. But readers are likely to think about it and wonder whether it undermines the authors' conclusions. Thus, explicitly addressing this argument, and whether it is qualitatively or quantitatively incorrect, would strengthen the manuscript.

2b) A related concern is that, if we understand correctly, the model produces only step-function psychometric curves: for a given stimulus *s*, the response is deterministic, and when *V_R_* = *V_L_*, the model would produce 100% correct behavior. This might seem unimportant, since the details of the shapes of psychometric curves are not the focus of the manuscript, and replacing the distribution-based model with one that uses noisy samples might seem a trivial change. However, distinguishing between hard and easy trials is central to the arguments in the manuscript, and the corresponding difference in error rates might easily become important (as in the situation sketched in point 2a above). So a model that produces smooth psychometric functions (as opposed to step-shaped) might be important after all (and at least cosmetically would be an obviously better match to the data).

2c) The authors posit that the subjects use an internal measure of decision confidence to update their decision policy. In support of this claim, they show that a qualitatively similar modulation can be produced by a temporal difference reinforcement learning (TDRL) model with prediction errors based on perceived stimulus strength. In this model, the prediction errors are used to update the stored value of each choice. We found this model design counterintuitive. It seems that an equivalent model could be constructed in which prior stimulus probabilities (*p_R_* and *p_L_*) are updated instead. This would be more consistent with the fact that the animal always receives the same amount of reward on rewarded trials, but may have uncertainty about exactly where to set the decision boundary leading to trial-to-trial updating. It would be helpful if the authors reformulated their model, or explained why reward values are being updated rather than reward probabilities.

2d) The Bayesian model that updates stimulus statistics seems to ignore which choice was made in the previous trial or whether it was correct. That is, there is a built-in handicap of no feedback, compared to the confidence-based models. How much does this "handicap" contribute to the poor match between model predictions and data in Figure 9A?

2e) Model predictions should be comparable to real data. In Figure 2, the plots for rats' average performance show three levels of previous stim (%A) for each direction, more or less evenly distributed. In Figure 3, however, previous stim (%A) are 20, 50 and another value very close to 50. Does the model performance depend on the stimulus strength used? The same stimulus strengths used in behavioral testing should be used for model predictions.

2f) Parameter values used to generate Figure 3 should be reported.

3) More detail and clarity are needed regarding the description of the basic phenomenon (e.g., sorting and analysis of the shown quantities). The Materials and methods section should include a brief section on how the psychophysical results were generated. That could include all the details of what was conditioned on what, as well as how the trials were divided into "Hard" and "Easy".

Many of the key figures depict "updating% " and "updating index." These terms should also be mathematically defined in the Materials and methods.

The equation used to fit the bias, lapse, sensitivity of psychometric curves should also be presented in the Materials and methods. These parameters are said to be "stable." What is the criterion for determining stability?

Also, the trial nomenclature and labels used in Figure 2 (e.g., "Next" and "Previous") were confusing.

Do subjects also show adjustments of sensitivity after correct trials? Would a change in sensitivity contribute to the observed confidence dependence (e.g., if the psychometric curves are shallower, the difference in choice might appear larger for difficult trials)?

4) What happens after errors? The results demonstrate an effect akin to a win-stay strategy but limited to "guesses" only. Is there also a trend toward the corresponding lose-switch strategy? That is, when the choice in trial n-1 is difficult and not rewarded, is the subject more likely to choose the *alternative* option in trial n (when such decision is also difficult)? There is no obvious reason to expect that confidence-guided choice updating would not also happen after errors, but in any case, error trials should not simply be put aside. The authors should present an analysis of error trials and if they do not see the effect predicted by the model, should propose an explanation for the inconsistency. Other work from the same first author (Lak et al., 2017), presented an alternative TDRL model without the "belief state," which seemed to have qualitatively similar results for correct trials but divergent results for errors. It seems that error trials are needed again in order to rule out this other model.

5) Related to this: "The choice updating remained statistically significant even after this correction." True, but the effect did seem to get a bit weaker. I imagine this is because the history effects are not limited to one trial in the past, but possibly more. If so, you would expect the effect to become stronger when the current difficult choice is preceded by two rewarded guesses made in the same direction. Whether the trend is weak or not, it could be compared to the model's prediction.

6) The transfer of choice updating across modalities is interesting. Comparing Figure 2, 4C, and 8D, it seems that the updating is larger on pure olfactory tasks and about the same for the pure auditory or mixed tasks (this is more obvious in Figure 10). I assume the rats have different sensitivity to olfactory and auditory stimuli. Then the difference seems to contradict the statement that "updating is guided by outcome expectations, rather than stimulus statistics". How much updating was there for trials in the mixed modality task but without modality switches?

7) Figure 10B and C should show scatterplots separately for each task. The authors' hypothesis is that updating is independent of stimulus statistics. If this is true, it makes sense to pool data across tasks. However, if the alternative is true that updating depends on certain properties of sensory stimulus, i.e., there could be different relationships between updating and slope/lapse, which could be obscured by pooling across tasks. A more direct test would be to fit the model to one task and predict results for the other tasks.

8) In Introduction and Discussion, the authors seem to suggest that this phenomenon persists after training is completed or after the subjects performed the task for extended periods and thus may reflect some optimal strategy. Was the training specifically targeting the suboptimal bias? Or is it possible that the subjects just settled on a suboptimal strategy that satisfies the training requirements? It might be useful to clarify what criteria were used to deem these subjects "well-learned".

[Editors' note: further revisions were suggested prior to acceptance, as described below.]

Thank you for resubmitting your article "Confidence-guided updating of choice bias during perceptual decisions is a widespread behavioral phenomenon" for consideration by *eLife*. Your revised article has been reviewed by three peer reviewers, including Emilio Salinas as the Reviewing Editor and Reviewer #1, and the evaluation has been overseen by Michael Frank as the Senior Editor. The following individuals involved in review of your submission have agreed to reveal their identity: Carlos D Brody (Reviewer #2); Long Ding (Reviewer #3).

The reviewers have discussed the reviews with one another and the Reviewing Editor has drafted this decision to help you prepare a revised submission.

Summary:

This manuscript presents interesting, important data characterizing how past choice history influences decisions that, in principle, should be guided just by currently available sensory information. The results show that the success of prior choices and the quality of the perceptual information guiding them are automatically tracked, and that this information is used to bias upcoming choices that are difficult, i.e., for which the quality of the sensory information is poor and the subject is largely guessing. The work is significant because the phenomenon seems robust across sensory modalities, species, task details, and labs, and because the modeling results provide insight into the underlying associative mechanisms responsible for the biasing effects.

Essential revisions:

The paper is significantly improved, and the reviewers are now convinced that it contains important work that should be published. Nevertheless, the presentation of the data is still confusing, and some further clarifications would substantially strengthen the manuscript.

1) Explaining more in detail the fundamental difference between the standard TDRL model and the belief model would be important. Specifically, the updating that they generate during easy trials, and how that updating depends on reward, could be the critical reason why the standard TDRL model fails.

2) The control analysis that corrects for potential slow drifts in the internal categorical boundary may also be implicitly addressing a separate issue (whether the shifts in the psychometric curve depend on the previous trial outcome, i.e., rewarded vs. not rewarded). Making this distinction explicit would be helpful.

3) The zig-zag patterns of the main data plots can be confusing because sometimes it is unclear what matters, the slope, the discontinuity, or both. Some suggestions for modifying the plots are provided below. That and/or additional text to guide the reader to the relevant features of the data would be helpful.

4) Figure 4 is a very nice addition to the manuscript. However, the results do differ somewhat from the data and from those of the belief model. Does this simply reflect a fundamental, qualitative difference between the classifier and the other models? It would be helpful to clarify whether the classifier has parameters that can be adjusted.

Details about each of these points are provided below.

Reviewer #2:

1) Figure 3 TDRL model: In the previous round of review, we (the reviewers) presented an argument as to why the TDRL model would lead to updating that depended on previous trial difficulty. The authors replied that this model results "in updating which is independent of previous difficulty." I believe that response is simply not true: in Figure 3C left, one can clearly see a non-zero slope to the updating% on hard trials. (And Figure 3C right worries me no end: since in the TDRL model there are no slow side biases independent of outcome, I am unsettled by the correction for such biases changing the results of the model.) [Note that doing the zig-zag plots in the style suggested below for Figure 1G would focus the eye on that slope much better.] What am I supposed to focus on? That the slope is less than in the belief model? That the belief model reaches zero updating for the easiest previous trials?

I spent a lot of time trying to think this one through – if the TDRL model could indeed be described as accounting for the data, that would be a pretty bad hit at the heart of the manuscript – and eventually hit upon something that I think could help the authors. Assuming I'm not getting this wrong, perhaps the authors had already thought of this, but either way, the suggestion is that it might be clarifying to have it in the manuscript. Here's the idea: in the TDRL model, *V_L_* and *V_R_* converge onto the average reward given when the subject chooses those ports, where the average is over both correct and incorrect trials. In other words, *V_L_* and *V_R_* are the value of the port, averaged over all trials. But in the belief model, *V_L_* and *V_R_* converge onto the value of the reward *when a reward is given*. Not an average over all trials, but conditioned on the reward having been given. Thus if the reward *r* = 1, in the belief model, *V_L_* and *V_R_* will converge on 1. And that means that for very easy trials, the reward prediction error will be zero, and the updating will be zero. In contrast, in TDRL they will converge on something like 0.8; and thus even on very easy correct trials there will be a non-zero RPE and non-zero updating. I may have gotten this wrong, but if it is correct, there are two interesting things here: (a) the difference on what *V_L_* and *V_R_* converge to in the two cases, in the sense of one being reward averaged over errors and corrects, the other being average reward conditioned on trials being correct; (b) I believe the real difference between TDRL and belief is not that TDRL has a zero slope for updating versus previous stimulus (the non-zero slope is right there in Figure 3C). It is that TDRL will never have zero updating for the easiest stimuli, whereas the belief model will.

While on this topic: don't Figure 9E and 9H look more like TDRL in Figure 3 than like the belief model? Why are they being interpreted as supporting the belief model?

2) Subsection “Choice updating is not due to slow drift in choice side bias”: This confused me the second, the third, *and* the fourth time I read it. Eventually I realized that there may be two issues being treated simultaneously here. I think things would be a lot clearer if you separated them. Issue (a) is "are shifts in the psychometric curve contingent on the previous trial's outcome?" Issue (b) is "are there slow drifts in the decision boundary that would induce correlations across trials that would make one trial appear to depend on the previous one"? The thing that confused me is that to solve issue (a), the obvious and easiest thing is to compare two psychometric curves, both conditioned on a previous stimulus *p*, according to the previous trial's outcome, i.e., whether the previous trial was rewarded or not. That would be a really easy plot to make, understand, and interpret: if previous trial's outcome matters, it will be obvious. Why not add it to the paper?

Issue (b) is also interesting. The approach in Figure 2 addresses issue (b). If this section and figure were described as focused on issue (b), it would be a lot easier to understand.

3) Figure 3 model: the full model really needs to be fully explained, in the main text. Please use equations. In particular, while the sentences “Note that although the choice computation is deterministic, the same stimulus can produce left or right choices caused by fluctuations in the percept due to randomized trial-to-trial variation around the stimulus identity (Figure 3—figure supplement 1)” is a welcome addition, it is not enough. Please specify, in the main text, how *p_R_* and *p_L_* are computed. Note that the integral in subsection “TDRL model with stimulus belief state” needs to specify what you're integrating with respect to (it needs a "ds"), this would make it clear that *p_R_* is a function of ŝ, which is itself a random variable drawn anew on each trial. I suggest that you make this explicit in the equation, by writing the left-hand-side as *p_R_(ŝ)*. (Note that I'm suggesting you bring this integral and some of the description into the main text.)

4) In the previous round, we requested substantial clarifications for panels of the type of Figure 1F and 1G. Even with the clarifications provided, I still find these panels hard to read.

– Figure 1F: this should be explained in a way that readers can understand without having to trawl through the Materials and methods. Here's my current understanding: (a) you plot the average psychometric curve; (b) you plot the psychometric curve conditioned on a particular previous trial stimulus *p*; (c) for each current stimulus *c*, you compute the vertical distance between those two curves, and that is what you call "updating".

Why not show this graphically directly, to make it easy for readers to understand? That is, something along the lines of: add a panel to Figure 1 where you show the average curve and the curve conditioned on *p*, add arrows pointing to the vertical differences between those two curves, and add an arrow from there to Figure 1F to indicate that these two particular curves and the vertical shift between them are what become column *p*Figure 1F.

Among other things, this would make it obvious why, if the psychometric curves asymptote at 0% and 100%, updating is necessarily going to be small for easy current stimuli. Which is why I don't like the plot of Figure 1F so much: the eye gets drawn to the dominant pattern, which is that the top and bottom row are lighter than the middle rows. But that's the unimportant part, that's the "expected" part, as the authors now write. The important part is happening in the middle rows. Could the authors think of a display format that focuses the eye on that, on the middle rows, instead of the already-expected parts?

– Figure 1G: The zig-zag pattern confused me no end. What is it that I'm supposed to focus on here? The difference between easy and hard? The fact that the pattern is antisymmetrical? The slope on the hard stimuli?

It eventually dawned on me that the "A" response and "B" response are of course anti-symmetrical with each other. For a model which has no intrinsic side bias, this has to be true. And there appears to be no systematic, overall, side bias across the experimental rat data in the paper. So the zig and the zag are actually redundant with each other.

I would therefore suggest the following: collapse the two with each other (the zig and the zag), which gives you better statistics, and in addition focuses the eye on the important parts, not the antisymmetry. In other words, instead of plotting as a function of previous odor A% , plot (% updating towards correct side) as a function of |A% – 50|. By halving the x and y axes, that would also allow you to zoom in by 2x, so readers can see the data better. (An added suggestion would be to plot the easy trials in a light grey, to emphasize that it's in the hard trials that the action is.) And then you'd have a plot that, at a single glance, tells the reader "there is bigger updating for harder stimuli".

Reviewer #3:

The authors addressed my earlier concerns. But their new data raised a new concern:

I think it is good of the authors to try another class of models to explain the data. However, the predictions of the statistical classifier in Figure 4D differ from experimental data in Figure 1G in three aspects: (1) there appears to be a strong dependence on previous stimul us strength for easy choices; (2) perhaps as a consequence, there is a large jump in Updating% going from green to blue at% A = 50; and (3) the range of updating% is only about half of the experimental data. It is hard to judge if these differences represent fundamental deficits of the model or just wrong parameterization. Because this model is not as intuitive as the RL model in Figure 3B, it would be helpful if the authors can expand this section (or add a supplemental figure) to give the readers a sense of how varying each model parameter changes the predictions.

---

## [Author Response]

Essential revisions:1) The first major concern is in regard to a claim made throughout the paper, namely that how easy or hard the current stimulus is affects the strength of the modulation. This is first presented in Figures 1F and 1G, and then repeated in almost all the figures. However, the measure presented in figures of the style of Figure 1F and summarized in Figure 1G is the% change in responses (given a particular current and previous stimulus, and relative to the overall average for that stimulus). This measure suffers from a large problem that, unfortunately, appears to not be addressed (apologies if it was missed on reading): there is a ceiling effect, in the sense that when a choice is easy (near 100%) it cannot increase further, even if there were strong underlying plasticity. The ceiling applies preferentially to easy stimuli, and therefore will manifest as a difference between easy and hard stimuli. It is currently impossible to tell whether the data shown in support of the claim of "current stimulus strength modulates effect" is merely an artifact resulting from this ceiling issue, or an actual finding. It is very important that the authors clarify this issue.

We agree with the reviewers that our core finding is the dependence of choices on the difficulty of previous trial. Indeed, this can be observed only when the “current” stimulus is difficult for the reason described by the reviewer. Psychometric curves have this property that bias appears as a shift of the curve with minimal effects for easy trials. In our revised manuscript we have made this point clear. We have clarified that when the current sensory evidence is strong it determines the choice independent of past trials. This is as expected and has been shown previously. However, when the current sensory evidence is weak, choices depend on the difficulty of previous stimuli (paragraph four in subsection “Perceptual decisions are systematically updated by past rewards and past sensory stimuli” and Discussion paragraph one).

Note that the absence of the effect of past trials when the “current” sensory evidence is strong is fully captured by our models. In the RL model choices are computed by comparing the expected vale of *L* and *R* actions (*Q_L_* and *Q_R_*). Because *Q_L_ = P_L_.V_L_* and *Q_R_* = *P_R_.V_R_* (the product of sensory confidence and learned reward), when current sensory evidence in favor of the choice is strong (large *P*), then choice is largely determined by *P* and the contribution of past trials (stored in *V*) is minimal, hence there is no bias from past trials when current sensory evidence is strong.

2a) Another key claim is that only the model with a belief state could account for the results. But the data are not convincing. Let's start with a TD model, without learning of actions, merely learning the values V_R_ and V_L_, and let's suppose the value of V_R_ is modified only after R choices (and similarly for V_L_ and L choices). Under these conditions (and assuming R-L symmetry) V_R_ and V_L_ will converge to the overall% correct. This will be more than 50%; for a typical smooth psychometric curve it might be 75% or 80%. Now move to a TDRL model that also learns actions: because there are more errors for hard stimuli, the error signal (V_C_ – r) will, on average, be greater for hard stimuli than for easy stimuli. And therefore, in this situation, there will on average be greater plasticity after hard stimuli than after easy stimuli, entirely without a belief state. Could this not account for the authors' experimental data?The situation and conclusion sketched here could be wrong. But readers are likely to think about it and wonder whether it undermines the authors' conclusions. Thus, explicitly addressing this argument, and whether it is qualitatively or quantitatively incorrect, would strengthen the manuscript.

In our revision we attempted to clarify this issue. We explain why models of the type described by the reviewers cannot fully capture the behavioral effect.

Although intuitively we agree that this argument appears compelling, in fact the suggested model does not account for the effect that we observed. Figure 3C shows the model that the reviewers described; it shows a model that computes prediction error by comparing outcome and average expectation (i.e. *V_L_* and *V_R_* converged over trials). The prediction errors are independent of the difficulty of previous decision, resulting in updating which is independent of previous difficulty. In this model with two states (*L* and *R*), the value functions (*V_L_* and *V_R_*) indeed converge over trials to reflect average performance (say 80%): *V_L_* = 80% and *V_R_* = 80%. For choice computations one can take two approaches: (1) entirely based on the sensory evidence, i.e. by comparing *P_L_* and *P_R_*, which will result in choices that are independent of past trials and hence it will not account for our observations; (2) comparison of *Q* values, *Q_L_ = P_L_ × V_L_* and *Q_R_ = P_R_ × V_R_*, which allows past trials to have an influence on the current choice. After the choice, prediction error is the difference between reward *r* and average performance (converged over trials). Consider two different situations: when the previous trial was a leftward rewarded trial and it was either easy or difficult. In both cases, the prediction errors (*r – V_L_*) are similar, and they hence update *V_L_* in a similar level. This would thus cause a choice bias in the next trial to be largely independent of the past sensory difficulty (Figure 3C). The model with two state produces confidence-dependent learning only if it includes the belief that each state is occupied in the current trial (i.e. the main model shown in Figure 3A, B).

We can also consider alternative models that produce different average expectations for different stimuli by representing stimuli across multiple states (rather than the two state version considered above). Such model learns the average value for each stimulus, i.e. the state values are converging to reflect the average value of each level of sensory evidence. In this model, in each trial is based on the internal sensory stimulus the model infers and assigns a single state to the stimulus. Prediction error could be computed in two ways: either it does not have access to this inferred state or it has the access to the inferred state. The first scenario makes prediction errors independent of past difficulty. The second scenario has access to the inferred state and compares reward with the value of that state to produce confidence- dependent prediction errors. However, since the updates only impact the current state there is no bias expected for different (nearby) stimuli.

Note that one of our observations, the transfer of updating across sensory modalities, also constrains our models. Choice updating is transferred between modalities and hence models that include multiple states to store the value of each stimulus cannot account for this behavioral observation. Conversely, the model with two states stores past values into the left and right *action* value and can account for updating across modalities. Based on this observation we suggest that the reduced model with two states is more appropriate for our findings.

We apologize that our description was not sufficiently clear and did not clearly convey the model illustrated in Figure 3C and did not spell out the necessity of the belief state. In the revised manuscript we have substantially expand on this issue in the text (Subsections “Belief-based reinforcement learning models account for choice updating” and “TDRL models without stimulus belief state”).

2b) A related concern is that, if we understand correctly, the model produces only step-function psychometric curves: for a given stimulus s, the response is deterministic, and when V_R_ = V_L_, the model would produce 100% correct behavior. This might seem unimportant, since the details of the shapes of psychometric curves are not the focus of the manuscript, and replacing the distribution-based model with one that uses noisy samples might seem a trivial change. However, distinguishing between hard and easy trials is central to the arguments in the manuscript, and the corresponding difference in error rates might easily become important (as in the situation sketched in point (2a) above). So a model that produces smooth psychometric functions (as opposed to step-shaped) might be important after all (and at least cosmetically would be an obviously better match to the data).

We have clarified that due to sensory noise even a deterministic choice rule will produce smooth psychometric curves (subsection “Belief-based reinforcement learning models account for choice updating” paragraph three). We have also clarified that using a softmax rule would not substantially change our results. To illustrate this point, we also added a supplemental figure (Figure 3—figure supplement 1).

The choice computation is entirely deterministic. However, since the percept is drawn from a normal distribution in each trial, the same stimulus can produce left or right choices across trials. For instance, an external stimulus (*s* = 55%) could result in an internal percept (*ŝ* = 40%) in one trial and 57% in the next trial leading to left and right choices respectively (assuming that 50% is the decision boundary). Thus, when averaging across trials, the psychometric curve will be graded and not step-like. We now show that using a non-deterministic, e.g. softmax, rule for choice computation does not influence our core findings (Figure 3—figure supplement 1).

2c) The authors posit that the subjects use an internal measure of decision confidence to update their decision policy. In support of this claim, they show that a qualitatively similar modulation can be produced by a temporal difference reinforcement learning (TDRL) model with prediction errors based on perceived stimulus strength. In this model, the prediction errors are used to update the stored value of each choice. We found this model design counterintuitive. It seems that an equivalent model could be constructed in which prior stimulus probabilities (p_R_ and p_L_) are updated instead. This would be more consistent with the fact that the animal always receives the same amount of reward on rewarded trials, but may have uncertainty about exactly where to set the decision boundary leading to trial-to-trial updating. It would be helpful if the authors reformulated their model, or explained why reward values are being updated rather than reward probabilities.

The class of models which update the probabilities or the position of the decision boundary indeed accounts for our data. In fact, we considered including such models in our initial submission but decided to leave it out to avoid having too many models. We agree that it is valuable to reformulate our findings using such models, and we have added one such model, a statistical classifier, to the revised manuscript (new Figure 4 and accompanying text in subsection “On-line learning in margin-based classifiers explains choice updating”).

Note that we did not claim that the model presented is the only one that accounts for our observations. Instead we conclude that several classes of model could account for our data, as long as they adjust their choice strategy based on the confidence in receiving reward in the preceding trial. However, there is one observation that favors our RL model (which updates action value) over models that update sensory probabilities. We observed that learning transfers across trials even when the evidence comes from different sensory modalities.

This question taps into a fundamental and important issue: which classes of models can account for our behavioral observations? It is clear that a purely sensory-based model or a purely reward-based model cannot account for the data. Therefore, we can start with a sensory-based model and make modifications so that decision boundary is adjusted according to past trials (reviewers’ suggestion), or to start with a reward-based model and modify it so that it considers sensory uncertainty (i.e. the teaching signal reflects past confidence as in our RL model). While various classes of models account for our data, they share one main computation: they adjust the degree of *learning based on the statistical confidence* in the accuracy of previous decisions. Bayesian learning in drift-diffusion models of decision making also makes similar predictions about confidence-dependent choice biases (Drugowitsch et al., 2019).

In our revised manuscript we have added an on-line Bayesian support vector machine model as a main figure (Figure 4) to show another class of models that could account for our data. This class of model is analogous to the ones suggested by the reviewers: it updates the decision boundary. We would like to keep our RL model as the main model as it might be easier for readers to understand the nature of learning in this class of models, and it also accounts for cross-modality effect in a simple manner.

We have also included a paragraph in the Discussionto clarify that a large class of models could account for our observations, as long as they include a trial-by-trial adjustment of choice strategy which scales by the confidence in obtaining the reward (subsection “Rewards induce choices bias in perceptual decisions”).

2d) The Bayesian model that updates stimulus statistics seems to ignore which choice was made in the previous trial or whether it was correct. That is, there is a built-in handicap of no feedback, compared to the confidence-based models. How much does this "handicap" contribute to the poor match between model predictions and data in Figure 9A?

We agree that this specific model ignores trial choice and reward and only updates stimulus statistics. We have added a new Bayesian model which includes these and does account for the updating effect (Figure 4). As such, in our revised manuscript we no longer present the model that reviewers questioned, and have removed the corresponding figure from the manuscript.

2e) Model predictions should be comparable to real data. In Figure 2, the plots for rats' average performance show three levels of previous stim (%A) for each direction, more or less evenly distributed. In Figure 3, however, previous stim (%A) are 20, 50 and another value very close to 50. Does the model performance depend on the stimulus strength used? The same stimulus strengths used in behavioral testing should be used for model predictions.

Model predictions do not depend on the stimulus levels used in the simulation. In our revised manuscript, we have plotted model prediction using 10 evenly distributed levels of stimuli (Figure 3). In general, we note that the exact value of stimuli in the model or in the behavioral data do not matter as long as they produce psychometric curves with graded levels of choice accuracy.

2f) Parameter values used to generate Figure 3 should be reported.

Parameter values are now reported in the text as well as the caption to Figure 3 and Figure 11. Thank you.

3) More detail and clarity are needed regarding the description of the basic phenomenon (e.g., sorting and analysis of the shown quantities). The Materials and methods section should include a brief section on how the psychophysical results were generated. That could include all the details of what was conditioned on what, as well as how the trials were divided into "Hard" and "Easy".Many of the key figures depict "updating% " and "updating index." These terms should also be mathematically defined in the Materials and methods.The equation used to fit the bias, lapse, sensitivity of psychometric curves should also be presented in the Materials and methods. These parameters are said to be "stable." What is the criterion for determining stability?Also, the trial nomenclature and labels used in Figure 2 (e.g., "Next" and "Previous") were confusing.

We have included this information as requested. We have devoted a subsection in the Materials and methodsto describe how psychometric plots are generated. We also defined the calculation of updating index and included equations for psychometric fitting. These are included in subsection “Data analysis and psychometric fitting”. We also included a new Figure (Figure 2—figure supplement 1) to visualize our psychometric measures and normalizations.

Do subjects also show adjustments of sensitivity after correct trials? Would a change in sensitivity contribute to the observed confidence dependence (e.g., if the psychometric curves are shallower, the difference in choice might appear larger for difficult trials)?

We have included an analysis on the effect of past sensory difficulty on the psychometric sensitivity (slope) in paragraph four of subsection “Perceptual decisions are systematically updated by past rewards and past sensory stimuli”. We did not observe any significant effect of past confidence on the psychometric slopes.

4) What happens after errors? The results demonstrate an effect akin to a win-stay strategy but limited to "guesses" only. Is there also a trend toward the corresponding lose-switch strategy? That is, when the choice in trial n-1 is difficult and not rewarded, is the subject more likely to choose the alternative option in trial n (when such decision is also difficult)? There is no obvious reason to expect that confidence-guided choice updating would not also happen after errors, but in any case, error trials should not simply be put aside. The authors should present an analysis of error trials and if they do not see the effect predicted by the model, should propose an explanation for the inconsistency. Other work from the same first author (Lak et al., 2017), presented an alternative TDRL model without the "belief state," which seemed to have qualitatively similar results for correct trials but divergent results for errors. It seems that error trials are needed again in order to rule out this other model.

In the revised manuscript we dedicate a small section (“Different strategies for choice updating after error trials due to different noise sources”) and a new figure (Figure 11) to updating effect after error trials. Note that error trials are few, compared to correct trials, making it hard to make firm conclusions.

The challenge is the diversity of such post-error effects. To paraphrase Leo Tolstoy’s famous opening sentence of the novel Anna Karenina: all correct trials are alike; each incorrect trial is incorrect in its own way. Correct perceptual performance requires appropriate processing and evaluation of the stimulus. In contrast there are many processes that can lead to incorrect performance without consideration of the stimulus, from inattention to lack of motivation to exploration. Indeed, in our behavioral data, the post-error behavioral effects are diverse, usually even within the same data set.

Note that the model’s predictions about post-correct trials is largely independent of the model parameters (i.e. learning rate, sensory noise, possible memory/decision noise). In contrast, the predictions of the model for post-error trials depend on which parameters are the dominant source of noise, i.e. whether errors are related to perception, or they are internal and due to memory of values. Briefly, there are parameter settings such that choice noise is dominated by perception, these errors cannot be systematically corrected hence there is no net updating (Figure 11). Alternatively, when there is internal noise, such as high learning rate, systematic updating patterns are observed (Figure 11).

We find that some individual subjects match these specific patterns (Figure 11) but in fact the diversity of post-error updating goes beyond these two patterns as expected. We believe this is the consequence of the modeling framework that does not take many processes into account that could lead to errors, such as attentional lapses, lack of motivation or exploration. These are also briefly discussed in subsection “All correct trials are alike; each incorrect trial is incorrect in its own way”.

5) Related to this: "The choice updating remained statistically significant even after this correction." True, but the effect did seem to get a bit weaker. I imagine this is because the history effects are not limited to one trial in the past, but possibly more. If so, you would expect the effect to become stronger when the current difficult choice is preceded by two rewarded guesses made in the same direction. Whether the trend is weak or not, it could be compared to the model's prediction.

We thank the reviewers for this suggestion. We have performed this analysis and compared with the model prediction. Figure 3—figure supplement 2 shows this analysis. As the reviewers guessed, the effect is slightly stronger after 2 rewarded low-confidence trials in the same direction, consistent with the model. This has been added to paragraph three of subsection “Belief-based reinforcement learning models account for choice updating”.

6) The transfer of choice updating across modalities is interesting. Comparing Figure 2, 4C, and 8D, it seems that the updating is larger on pure olfactory tasks and about the same for the pure auditory or mixed tasks (this is more obvious in Figure 10). I assume the rats have different sensitivity to olfactory and auditory stimuli. Then the difference seems to contradict the statement that "updating is guided by outcome expectations, rather than stimulus statistics". How much updating was there for trials in the mixed modality task but without modality switches?

We thank the reviewer for this question; indeed, the data suggest that there is outcome-based updating in the space of action values but also that it is not the entire story. In our revision we have included the updating effect without modality switch in the task with mixed modalities (Figure 9—figure supplement 1). Updating appears largest in consecutive olfactory trials but it exists in both within and across modality switches.

We have added the following paragraph to the Discussionto discuss this point:

“Trial-by-trial transfer of choice updating across sensory modalities provides some evidence that this form of learning is driven by comparing the decision outcome with confidence – dependent expectation and performing updating in the space of action values. Updating across trials with different modalities appeared relatively smaller than trials without the switch in the sensory modality. This observation might point to the fact that in trial-by-trial learning animals follow a mixture of two strategies: one which updates values in the space of actions, and one that keeps track of stimulus identity and statistics across trials for such learning. The trade-off between these model-free and model-based trial-by-trial learning during perceptual decisions remains to be explored in future studies.”

7) Figure 10B and C should show scatterplots separately for each task. The authors' hypothesis is that updating is independent of stimulus statistics. If this is true, it makes sense to pool data across tasks. However, if the alternative is true that updating depends on certain properties of sensory stimulus, i.e., there could be different relationships between updating and slope/lapse, which could be obscured by pooling across tasks. A more direct test would be to fit the model to one task and predict results for the other tasks.

In our revised manuscript we have separately plotted the relationship between the size of confidence-dependent updating and psychometric slope/lapse for each dataset, as suggested by reviewers (new Figure 10).

We would like to clarify that our aim in this analysis is to test if the updating is stronger/weaker in individuals with better psychometric curves (i.e. higher slope and lower lapse). We find a mild inverse relationship between lapse rate and the degree of updating. This suggests that updating is not due to a lack of understanding of the task. The aim of this analysis was not to test the relation between stimulus statistics and updating.

8) In Introduction and Discussion, the authors seem to suggest that this phenomenon persists after training is completed or after the subjects performed the task for extended periods and thus may reflect some optimal strategy. Was the training specifically targeting the suboptimal bias? Or is it possible that the subjects just settled on a suboptimal strategy that satisfies the training requirements? It might be useful to clarify what criteria were used to deem these subjects "well-learned".

The training did not target any specific bias: both animals and human observers were presented with random sequences of trials and were rewarded for correct choices. The goal of training in each dataset was to achieve high-quality and stable psychometric performance. We consider the subjects to be well-trained because the psychometric fitting parameters reported for each dataset show asymptotic and stable behavior (i.e. sharp psychometric slope, minimal overall bias and near zero lapse). These have been reported for each dataset tested. We have also clarified in the Materials and methodsthat the training did not target or promote confidence- dependent bias (subsection “Rat olfactory experiment”).

[Editors' note: further revisions were suggested prior to acceptance, as described below.]

Essential revisions:The paper is significantly improved, and the reviewers are now convinced that it contains important work that should be published. Nevertheless, the presentation of the data is still confusing, and some further clarifications would substantially strengthen the manuscript.1) Explaining more in detail the fundamental difference between the standard TDRL model and the belief model would be important. Specifically, the updating that they generate during easy trials, and how that updating depends on reward, could be the critical reason why the standard TDRL model fails.

We have expanded on this issue and clarified the differences between the models by adding new figures and text (see below for the detailed answer).

2) The control analysis that corrects for potential slow drifts in the internal categorical boundary may also be implicitly addressing a separate issue (whether the shifts in the psychometric curve depend on the previous trial outcome, i.e., rewarded vs. not rewarded). Making this distinction explicit would be helpful.

We have addressed this and followed the reviewer’s advice to clearly dissociate these two issues. We have added new analyses, figures and text (see below for the detailed answer).

3) The zig-zag patterns of the main data plots can be confusing because sometimes it is unclear what matters, the slope, the discontinuity, or both. Some suggestions for modifying the plots are provided below. That and/or additional text to guide the reader to the relevant features of the data would be helpful.

We appreciate the suggestions for modifying these plots. We have now added a new figure panel to explain the calculations underlying the plot, have changed the colors in the plots for clarification, and have included additional text to clarify the analyses (see below for the detailed answer).

4) Figure 4 is a very nice addition to the manuscript. However, the results do differ somewhat from the data and from those of the belief model. Does this simply reflect a fundamental, qualitative difference between the classifier and the other models? It would be helpful to clarify whether the classifier has parameters that can be adjusted.

The difference simply reflects model parameters. We have now corrected this and expanded on this section for further clarification (see below for the detailed answer).

Details about each of these points are provided below.Reviewer #2:1) Figure 3 TDRL model: In the previous round of review, we (the reviewers) presented an argument as to why the TDRL model would lead to updating that depended on previous trial difficulty. The authors replied that this model results "in updating which is independent of previous difficulty." (Response to reviewers.) I believe that response is simply not true: in Figure 3C left, one can clearly see a non-zero slope to the updating% on hard trials. (And Figure 3C right worries me no end: since in the TDRL model there are no slow side biases independent of outcome, I am unsettled by the correction for such biases changing the results of the model.) [Note that doing the zig-zag plots in the style suggested below for Figure 1G would focus the eye on that slope much better.] What am I supposed to focus on? That the slope is less than in the belief model? That the belief model reaches zero updating for the easiest previous trials?I spent a lot of time trying to think this one through, if the TDRL model could indeed be described as accounting for the data, that would be a pretty bad hit at the heart of the manuscript, and eventually hit upon something that I think could help the authors. Assuming I'm not getting this wrong, perhaps the authors had already thought of this, but either way, the suggestion is that it might be clarifying to have it in the manuscript. Here's the idea: in the TDRL model, V_L_ and V_R_ converge onto the average reward given when the subject chooses those ports, where the average is over both correct and incorrect trials. In other words, V_L_ and V_R_ are the value of the port, averaged over all trials. But in the belief model, V_L_ and V_R_ converge onto the value of the reward when a reward is given. Not an average over all trials, but conditioned on the reward having been given. Thus if the reward r = 1, in the belief model, V_L_ and V_R_ will converge on 1. And that means that for very easy trials, the reward prediction error will be zero, and the updating will be zero. In contrast, in TDRL they will converge on something like 0.8; and thus even on very easy correct trials there will be a non-zero RPE and non-zero updating. I may have gotten this wrong, but if it is correct, there are two interesting things here: (a) the difference on what V_L_ and V_R_ converge to in the two cases, in the sense of one being reward averaged over errors and corrects, the other being average reward conditioned on trials being correct; (b) I believe the real difference between TDRL and belief is not that TDRL has a zero slope for updating versus previous stimulus (the non-zero slope is right there in Figure 3C). It is that TDRL will never have zero updating for the easiest stimuli, whereas the belief model will.While on this topic: don't Figure 9E and 9H look more like TDRL in Figure 3 than like the belief model? Why are they being interpreted as supporting the belief model?

We appreciate the reviewer’s suggestions and have followed the advice. In short, we agree with the reviewer’s insights. There are two differences between the belief-based model and the TDRL. First, the values converge through learning to two different quantities: in the belief- based model they converge to just below the true size of reward (i.e. reward value, given the reward given) and in the TDRL they converge to the average performance (average rewards over trials). This has implications for the size of prediction error for the easiest correct choices, as the reviewer described. The second difference among models is that in the belief-based model the size of prediction errors scales with the difficulty of previous choice while in the TDRL prediction errors shows little scaling with choice difficulty (see below for further details on this). As such, regardless of the magnitude of converged values, the teaching signals in the two models differ, and the teaching signals in the belief-based model causes different levels of learning in the next trial. We have expanded on this and spelled out these differences in the final paragraph of subsection “Belief-based reinforcement learning models account for choice updating”. We have also added new figure panels (Figure 3—figure supplement 1C,D) to further describe these differences.

The two effects, small to no updating for perfect accuracy (typically the easiest stimuli) and increasing updating for lower accuracy choices are both consequences of the model, so we are not trying to focus on one compared to the other. Note that zero updating would be only expected for maximum confidence choices with zero lapse rates in our model and the easiest stimuli differ across data sets, often producing less than 100% accuracy, hence non-zero updating is expected. On the other hand, we agree that zero updating for zero lapse rate trials is a central point that likely hid the role of reinforcement learning in perceptual decisions.

Regarding weak effect of previous trial difficulty for the TDRL model (Figure 3C) that the reviewer pointed out, our point is not only that this effect is weak compared to the main model. Rather, this effect vanishes after our normalization (Figure 3C left). Consequently, after applying the normalization, the only model that still shows updating like our subjects is the belief-based model (Figure 3B, left). Also note that in all TDRL models there is some correlation across trials due to the stored values that are by definition correlated across trials (similar to a drifting boundary in the signal detection model (see below). Hence it is important to evaluate models after the normalization (righthand panels of Figure 3). These issues have been clarified in subsection “Belief-based reinforcement learning models account for choice updating”.

How could a correlation across trials cause small difficulty-dependent effects on the updating in the TDRL model before applying the normalization (Figure 3C left)? Let us first consider the prediction errors in this model after a correct choice. They are largely independent of the choice difficulty. In fact, if anything, they are even marginally smaller when getting a reward after a difficult choice (Figure 3—figure supplement 1D), i.e. opposite to the belief-based model. This pattern in the prediction errors is due to correlation of values across trials, and how stored values and current sensory evidence interact in computing choice (*Q = V*P*). The model has a better chance of getting a difficult stimulus correct if it happens that it is transiently in the regime that the stored value of the correct side (i.e. stimulus side) is higher than the stored value of the other side (hence a difference in *Q* values). In this situation, at the time of reward prediction error computation, the reward is compared with relatively high stored value of the chosen side, resulting in marginally smaller positive prediction error (Figure 3—figure supplement 1D). Consequently, after this trial the value difference between *L* and *R* still persists. Let’s now consider a next trial that happens to be difficult and correct (with similar stimulus/choice side to the previous trial). The model comes to this trial with the persisting value difference described above, and again has a good chance of getting the choice correct. This bias appears as an apparent updating. These value difference-induced effects are more likely to occur for difficult stimuli (in the case of easy stimuli, choices are dominated by the sensory stimulus), resulting in slightly larger updating for difficult stimuli and hence the weak effect in Figure 3C, left. These, and other similar considerations highlight the fact that the models should compared and evaluated after normalizations to minimize correlations across trials. These have been briefly described in subsections “Belief-based reinforcement learning models account for choice updating” and “TDRL models without stimulus belief state”.

Finally, the reviewer is correct that in different datasets we are seeing various levels of choice updating and those in Figure 9 show weaker effects compared to some of the other datasets. The effect for each individual subject has been quantified in Figure 10 indicating the strength of the effect across subjects. Despite the diversity, many subjects show a substantial level of choice updating even in the data of Figure 9. In addition, the dataset presented in Figure 9 is particularly interesting, because we show that while there is some level of choice updating that is transferred across sensory modalities, the strength of choice updating within each modality is somewhat stronger (Figure 9—figure supplement 1). We speculate that this observation might point to the fact that in trial-by-trial learning animals follow a mixture of two strategies: one which updates values in the space of actions, and one that keeps track of stimulus identity and statistics across trials for such learning. This was mentioned in the text.

2) Subsection “Choice updating is not due to slow drift in choice side bias”: This confused me the second, the third, and the fourth time I read it. Eventually I realized that there may be two issues being treated simultaneously here. I think things would be a lot clearer if you separated them. Issue (a) is "are shifts in the psychometric curve contingent on the previous trial's outcome?" Issue (b) is "are there slow drifts in the decision boundary that would induce correlations across trials that would make one trial appear to depend on the previous one"? The thing that confused me is that to solve issue (a), the obvious and easiest thing is to compare two psychometric curves, both conditioned on a previous stimulus p, according to the previous trial's outcome, i.e., whether the previous trial was rewarded or not. That would be a really easy plot to make, understand, and interpret: if previous trial's outcome matters, it will be obvious. Why not add it to the paper?Issue (b) is also interesting. The approach in Figure 2 addresses issue (b). If this section and figure were described as focused on issue (b), it would be a lot easier to understand.

We have followed the reviewer’s advice. Addressing issue (a), we have added an example figure that the reviewer has suggested (psychometric curve conditional on one previous stimulus separated based on previous outcome; Figure 1—figure supplement 1) and have added related text to the main text in subsection “Perceptual decisions are systematically updated by past rewards and past sensory stimuli”. This, together with extensive analyses we presented in Figure 1 (looking at post-correct trials only) demonstrate our main point, that is the effect of the previous reward on the next choice depends on the previous choice difficulty.

Addressing issue (b), we have extensively expanded the manuscript to explain the importance of considering correlation across trials. We agree that the normalization may not be entirely intuitive. Therefore, we include a new simulation to illustrate how slow boundary shift alone can produce apparent trial-to-trial updating that is captured by the normalization method Figure 2—figure supplement 1). We first show that a signal detection model simulation with a fixed boundary does not show any effect of previous trials, as expected (Figure 2—figure supplement 1A,B). We then show that if we use drifting boundary in the simulation it will result in an apparent updating in the data (Figure 2—figure supplement 1C). Lastly, we show that our normalization method can correct for this (Figure 2—figure supplement 1D). We hope that this clarifies how the normalization works. We have also added additional to text to explain this point (subsection “Choice updating is not due to slow drifts in choice side bias”).

3) Figure 3 model: the full model really needs to be fully explained, in the main text. Please use equations. In particular, while the sentence “Note that although the choice computation is deterministic, the same stimulus can produce left or right choices caused by fluctuations in the percept due to randomized trial-to-trial variation around the stimulus identity (Figure 3—figure supplement 1)” is a welcome addition, it is not enough. Please specify, in the main text, how p_R_ and p_L_ are computed. Note that the integral in subsection “TDRL model with stimulus belief state” needs to specify what you're integrating with respect to (it needs a "ds"), this would make it clear that p_R_ is a function of ŝ, which is itself a random variable drawn anew on each trial. I suggest that you make this explicit in the equation, by writing the left-hand-side as p_R_(ŝ). (Note that I'm suggesting you bring this integral and some of the description into the main text.)

We have added more details, including equations about the main model in the text, as suggested, and have clarified the computation of *p_R_* and *p_L_* in the main text (see subsection “Belief-based reinforcement learning models account for choice updating”). We have also corrected the typo in the integral equation. Thank you.

4) In the previous round, we requested substantial clarifications for panels of the type of Figure 1F and 1G. Even with the clarifications provided, I still find these panels hard to read.– Figure 1F: this should be explained in a way that readers can understand without having to trawl through the Materials and methods. Here's my current understanding: (a) you plot the average psychometric curve; (b) you plot the psychometric curve conditioned on a particular previous trial stimulus p; (c) for each current stimulus c, you compute the vertical distance between those two curves, and that is what you call "updating".Why not show this graphically directly, to make it easy for readers to understand? That is, something along the lines of: add a panel to Figure 1 where you show the average curve and the curve conditioned on p, add arrows pointing to the vertical differences between those two curves, and add an arrow from there to panel f to indicate that these two particular curves and the vertical shift between them are what become column p in the panel in Figure 1F.Among other things, this would make it obvious why, if the psychometric curves asymptote at 0% and 100%, updating is necessarily going to be small for easy current stimuli. Which is why I don't like the plot of Figure 1F so much: the eye gets drawn to the dominant pattern, which is that the top and bottom row are lighter than the middle rows. But that's the unimportant part, that's the "expected" part, as the authors now write. The important part is happening in the middle rows. Could the authors think of a display format that focuses the eye on that, on the middle rows, instead of the already-expected parts?

The reviewer’s understanding about the underlying calculations is completely correct. We have followed reviewer’s advice and added a graphical description of these calculations (Figure 1D) and we have also included additional sentences in the main text for further clarification (subsection “Perceptual decisions are systematically updated by past rewards and past sensory stimuli”).

– Figure 1G: The zig-zag pattern confused me no end. What is it that I'm supposed to focus on here? The difference between easy and hard? The fact that the pattern is antisymmetrical? The slope on the hard stimuli?It eventually dawned on me that the "A" response and "B" response are of course anti-symmetrical with each other. For a model which has no intrinsic side bias, this has to be true. And there appears to be no systematic, overall, side bias across the experimental rat data in the paper. So the zig and the zag are actually redundant with each other.I would therefore suggest the following: collapse the two with each other (the zig and the zag), which gives you better statistics, and in addition focuses the eye on the important parts, not the antisymmetry. In other words, instead of plotting as a function of previous odor A% , plot (% updating towards correct side) as a function of |A% – 50|. By halving the x and y axes, that would also allow you to zoom in by 2x, so readers can see the data better. (An added suggestion would be to plot the easy trials in a light grey, to emphasize that it's in the hard trials that the action is.) And then you'd have a plot that, at a single glance, tells the reader "there is bigger updating for harder stimuli". Right now the zig-zag makes me squint 40 times at it before I get it.

We appreciate reviewer’s suggestion to fold the zig-zag plots across the stimulus axis. However, we think that the unfolded zig-zag plot represents a more appropriate signature of the updating effect. As the reviewer pointed out, among other things, it reveals potential side bias in the data. Moreover, it shows the asymmetry of the effect, which is a strong tendency to repeat left action or right action for nearby stimuli (in the middle of the x-axis) that would be obscured after folding. We have, however, accommodated part of the reviewer’s suggestion and changed the color of easy trials to grey and difficult trials to black so that the attention can be focused on the difficult trials. Once again, we are most grateful for all the helpful advice. Also, please note that the subject-by-subject analyses we presented (Figure 10) is similar to the reviewer’s suggestion as in that figure we are folding the zig-zag plots and use those for statistics.

Reviewer #3:The authors addressed my earlier concerns. But their new data raised a new concern:I think it is good of the authors to try another class of models to explain the data. However, the predictions of the statistical classifier in Figure 4D differ from experimental data in Figure 1G in three aspects: (1) there appears to be a strong dependence on previous stimulus strength for easy choices; (2) perhaps as a consequence, there is a large jump in Updating% going from green to blue at% A = 50; and (3) the range of updating% is only about half of the experimental data. It is hard to judge if these differences represent fundamental deficits of the model or just wrong parameterization. Because this model is not as intuitive as the RL model in Figure 3B, it would be helpful if the authors can expand this section (or add a supplemental figure) to give the readers a sense of how varying each model parameter changes the predictions.

We thank the reviewer for pointing out this apparent inconsistency and apologize for our insufficient explanation. The differences arose from a few model parameters, i.e. the rate by which the weight vector is updated and the level of sensory noise in the stimuli. For these parameters, we previously used the quantities used in the main model. However, it is straightforward to achieve updating results much more similar to the behavioral data by adjusting these parameters. In our revised manuscript, we have corrected the figure to show the results of our latest simulation. We have included the value of parameters used and also expanded on the model description in the Materials and methods section.